# Genomic organization, domain assortments, and nucleotide-binding domain diversity of NLR proteins in *Sordariales* fungi

Lucas Bonometti[1], Florian Charriat[1], Noah Hensen[2], Silvia Miñana-Posada[1], Hanna Johannesson[2,3], Pierre Gladieux[1]*

1 Plant Health Institute Montpellier, University of Montpellier, INRAE, CIRAD, IRD, Institut Agro, Montpellier, France, 2 Department of Ecology, Environment and Plant Sciences, Stockholm University, Stockholm, Sweden, 3 The Royal Swedish Academy of Sciences, Stockholm, Sweden

* pierre.gladieux@inrae.fr

## Abstract

Fungi have NOD-Like receptors (NLRs), homologous to the innate immune receptors found in animals, plants and bacteria. Fungal NLRs are characterized by a great variability of domain organizations, but the identity of the nucleotide-binding domains, the genomic localization, and the factors associated with variation in the composition of repertoires of fungal NLRs are not yet fully understood. To better understand the variability of fungal NLR repertoires and the underlying determinants, we conducted a thorough analysis of genome data from the ascomycete order *Sordariales*. Using similarity searches based on hidden Markov models profiles for canonical N-terminal, nucleotide-binding, or C-terminal domains, we characterized 4613 NLRs in 82 *Sordariales* taxa. By examining the Helical Third section of the nucleotide-binding domains, we substantially improved their annotation. We demonstrated that fungi have NACHT domains of both NAIP-like and TLP1-like types, similar to animals. We found that the number of NLR genes was highly variable among *Sordariales* families, and independent of the stringency of defense mechanisms against genomic repeat elements. NLRs were organized in clusters in the majority of taxa, and the strong correlation between the number of NLRs and the number of NLR clusters suggested that organizing in clusters may contribute to repertoire diversification. Our work highlights the similarity of fungal and animal NLRs in terms of nucleotide-binding domain types, and between fungal and plant NLRs in terms of genomic organization in clusters. Our findings will aid in the comparative analysis of the patterns and processes of diversification of NLR repertoires in various lineages of fungi and between the different kingdoms and domains of life.

---

**Data availability statement:** Genome assemblies, gene models, HMM profiles, orthogroups and NLR sequences generated in this study are available from Zenodo (doi: 10.5281/zenodo.14204294). The Python script for filtering is available on our github (https://github.com/bonospora/NLR_Sordariales).

**Funding:** This work was supported by The Bergianus Foundation, The Royal Swedish Academy of Sciences, to HJ. The funders had no role in study design, data collection and analysis, decision to publish, or preparation of the manuscript.

**Competing interests:** The authors have declared that no competing interests exist.

## Author summary

Fungi have NOD-Like receptors (NLRs), homologous to the innate immune receptors found in animals, plants and bacteria. Fungal NLRs are highly diverse, and how they evolve or function remains unclear. To explore this, we analyzed the genomes of 82 species from the fungal group *Sordariales*. Using advanced similarity searches, we identified 4,613 NLRs and found that the number of NLR genes varied widely between species, but this variation was not linked to the efficiency of mechanisms of defense against gene duplication. NLR genes often appeared in clusters in fungal genomes, similar to patterns seen in plant NLRs, and organization in clusters may contribute to NLR diversification over time. Some fungal NLRs also shared key features with animal NLRs, including two main types of NACHT nucleotide-binding domains (NAIP-like and TLP1-like). Our findings reveal surprising similarities in immune-like systems across fungi, animals, and plants. This work helps us understand how NLRs diversify and evolve across different forms of life.

## Introduction

Nod-Like Receptors (NLRs) are intracellular immune receptors that trigger the innate immune response and control biotic interactions. NLRs are present across the tree of life, in plants and animals, fungi, as well as in bacteria and archaea [1–6] (Fig 1A and 1B). In bacteria, plants, and animals, NLRs initiate an innate immune response upon detecting pathogen-derived ligands or host molecules released after cellular stress or damage. In fungi, some NLRs are involved in conspecific non-self recognition (*i.e.*, vegetative incompatibility) [7], but could more generally represent components of an undiscovered fungal innate immune system [8,9]. Understanding NLR function and evolution across different kingdoms is crucial to disentangling the origin of immune systems [10], and developing new approaches to manipulate immune responses [11].

NLRs follow a modular tripartite structure, which includes a C-terminal domain, a Nucleotide Binding (NB) domain, and a N-terminal domain (Fig 1A). The C-terminal domain acts mostly as a ligand-binding platform with repeated domains. Upon binding of a ligand (*e.g.*, a virulence effector from a pathogen), the C-terminal domain activates the central Nucleotide Binding (NB) domain [6,20–22], which is a nucleoside triphosphatase of the STAND (Signal Transduction ATPases with Numerous Domains) superfamily [23]. STAND domains likely evolved from a common bacterial ancestor [1,23,24], and diversified in NB domains of NLRs into two different lineages, called NACHT (NAIP, CIIA, HET-E, TLP1) and NB-ARC (Nucleotide-Binding adaptor shared by APAF1, plant R proteins, and CED4). NB domains of fungal NLRs are either of the NACHT or NB-ARC type. The NB domain serves as an ADP-ATP switch [25] that mediates the activity of the NLR through oligomerization of the monomeric NLR with other, not necessarily identical, NLRs (Fig 1C; [6]). NLR oligomers can form complexes with other proteins [26–28]. NLR oligomerization triggers signal

PLOS Genetics

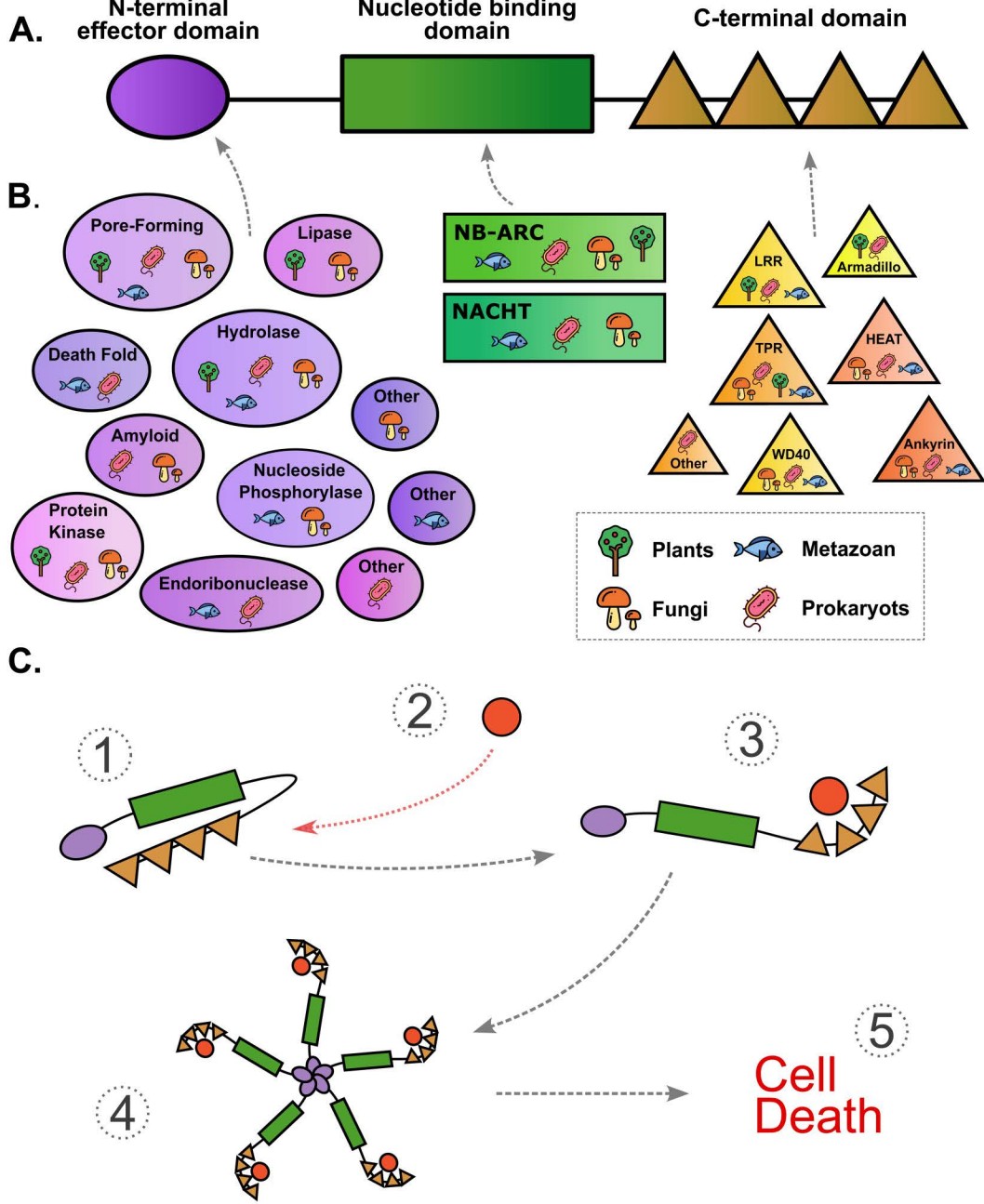

**Fig 1. Architecture of NLR proteins.** (A) Schematic representation of the domain architecture of an NLR, including an N-terminal effector domain, a central NB domain, and a C-terminal domain composed of repeated elements. (B) Most common functional or structural groups of constitutive domains of NLRs shared by plants (based on [12,13]), animals (based on [14,15]), fungi (based on [2,16]), and/or prokaryotes (based on [17,18]). In fungal NLRs, pore-forming domains correspond to HeLo-like and Goodbye-like domains; lipase domains correspond to SesB-like and patatin domains; hydrolase domains correspond to the HET-like annotation; nucleoside phosphorylase domains correspond to the PNP_UDP_1 domain; protein kinase domains correspond to the PKinase domain; amyloid domains correspond to the amyloid domain, while the "other" category includes, among others, the RelA_SpoT domain. (C) Schematic representation of protein mechanisms putatively involved in the activation of a fungal NLR, based on what is known in plants [19]. (1) NLR is in an inactive state, in which the C-terminal domain represses the NLR activation. (2) A specific ligand, such as an effector protein, binds the C-terminal domain and induces a conformational change in the NLR protein. (3) The NLR is in an active state. (4) Through their NB domains, individual NLRs oligomerize to form a wheel-like structure. (5) At the center of the structure, N-terminal domain assembly induces a gain of function that initiates an immune response. The type of response depends on the N-terminal domains but often leads to programmed cell death.

transduction, mediated by the N-terminal domain, also called the effector or signaling domain [29]. In fungi, Wojciechowski *et al.* [16] identified 27 N-terminal annotations, clustered in 17 domain families. A substantial fraction of fungal NLRs can also have amyloid motifs or unannotated short sequences as N-terminal domains.

NLR repertoires are highly variable in size, in plants [6,15], animals [14,30–34], and fungi [2,4]. Fungi and early-branching animals, unlike plants and mammals, also display high richness in the combinatorial associations of their various N-terminal, NB, and C-terminal domains [14,31]. NLRs are frequently found in clusters within genomes, both in plants [35] and animals [3], some clusters reaching several megabases in size. Aside from large clusters, plant NLRs can also be found in pairs, often in head-to-head arrangement, and these NLRs tend to be highly diverged [35]. In head-to-head pairs of plant NLRs, the sensing and signaling functions are uncoupled in two distinct proteins (the "sensor" NLR and the "helper" NLR; [36]), encoded by genes located close to each other, in inverted tandem arrangement, often sharing promoter [40,41].

To unlock the full potential of comparative immunological analyses between different kingdoms of life, several aspects of the evolution of fungal NLRs need to be clarified. Firstly, the factors associated with variation in NLR content remain unknown, as the repertoires of fungal NLRs have so far been characterized at the scale of *Dikarya* [2,4,16], the depth of which approaches 400 million years [37]. It is hypothesized that fungi in mycopathogen-rich environments will have more NLRs to detect and respond to a wide range of pathogens. However, mechanisms of genome defense against transposable elements, such as repeat-induced point mutation (RIP; [38]), should conflict with mechanisms of organismal defense based on the duplication of NLR receptors. Secondly, despite recent advances in the annotation of N-terminal domains [16], our knowledge of domain architecture remains fragmented. This is mainly because the NB domain often remains uncharacterized, as is the possible existence of NLRs with extra domains acting as decoys for virulence effectors [39] or head-to-head NLRs [40,41]. The rapid growth in the number of sequenced fungal genomes – notably in *Sordariomycetes* [42] – and the constant developments in the prediction of functional domains open up new opportunities to understand how the fungal repertoires of NLRs diversified.

Here, we report on the composition and genomic organization of NLR repertoires in *Sordariales* fungi (*Ascomycotina*). *Sordariales* fungi can be sampled on a range of substrates such as dung, wood, leaves, litter, burned vegetation, biological crusts, and soil [43,44], suggesting a wide variety of life cycles, and possibly differences in NLR content, if the ecology of organisms impacts the composition of their immune receptor repertoire. *Sordariales* encompass taxa such as *Neurospora* and *Podospora*, which are model systems for the study of conspecific non-self recognition systems that have been shown to involve NLRs [45,46]. Our study aimed to (i) precisely define the composition of NLR repertoires in *Sordariales*, (ii) investigate the existence of NLR clusters and NLRs arranged head-to-head, (iii) assess the relationship between the strength of mechanisms of genome defense against genomic repeat elements and the number of NLRs and NLR clusters, (iv) improve the annotation of NB domains, (v) determine the presence and distribution of *Sordariales* NLRs presenting extra domains in addition to the canonical N-terminal, NB, and C-terminal domains, which may act as integrated decoys against virulence effectors from mycopathogens.

## Results

### Dataset

We used a collection of genome assemblies that included 82 haploid isolates of *Sordariales* fungi and two outgroups representing 27 genera (S1 Table). Assembly size ranged from 28.3Mb to 57.7Mb, with an average size of 39.3Mb (standard deviation [s.d.]: 6.0Mb). L50 ranged from two to 704 contigs (mean: 83.8; s.d.: 128.9) and N50 from 14.1Kb to 11.2Mb (mean: 2.29Mb; s.d.: 2.82Mb; S1 Table). The completeness of assemblies, as estimated using Busco [47], ranged between 77.3 and 99.2% (mean: 96.1%; s.d.: 3.7%), and the number of predicted genes varied from 7,066–14,970 (mean: 10,582; s.d.: 1,788; S1 Table). Together, these observations indicate that our dataset is of sufficient quality to probe into the evolution of a family of genes such as NLRs.

## Species tree inference

To place our analyses in an evolutionary context, we constructed a species tree using two different species tree inference methods, based on single-copy orthologs. Orthology analysis of the 82 ingroup genomes and two outgroup genomes identified 77,480 orthogroups, of which 2367 orthogroups were single-copy orthogroups that were present in >80% of the genomes included in the dataset (referred to as SCO80), and 1030 orthogroups were single-copy orthogroups present in all genomes (referred to as SCCO). The species tree inferred using the gene tree summary method implemented in Astral-III [48] based on SCO80 was supported by high confidence values (Fig 2A). Local Posterior Probability (localPP) was maximal for all nodes, except the node including *Podospora bellae-mahoneyi* and *P. pseudocomata* (localPP: 0.92), and the node carrying the three *N. tetrasperma* lineages (localPP: 0.68 and 0.99). The species tree recovered the main *Sordariales* families as monophyletic groups, and confirmed the polyphyly of *Podospora*, *Cercophora,* and *Chaetomium*. The total-evidence species tree inferred using a maximum-likelihood approach based on the concatenation of SCCO sequences was mostly congruent with the Astral-III species tree (S1 Fig). Incongruence between Astral and total-evidence species trees was limited to the nodes with the lowest bootstrap support in the latter. All subsequent analyses were carried out using the Astral-III phylogram as the species tree, after having inferred branch lengths using the SCCO sequences (Fig 2A).

## High variability in NLR content across *Sordariales*

To investigate whether the extensive variability in NLR content observed at the scale of *Dikarya* is also observed at the more restricted scale of *Sordariales*, and to investigate if there are differences between families or within families of *Sordariales*, we used a combination of Pfam-A and *Sordariales*-specific HMM profiles which identified 4152 genes with an NB domain and canonical C-terminal repeats (Ankyrin, TPR, WD40, or HEAT; Fig 3). The methodology employed to predict NLR genes is summarized in Fig 3 and detailed in the Methods section. HMM profiles were generated from alignments of domains identified using Pfam-A, with the aim of improving detection sensibility in *Sordariales*.

The 4152 genes were clustered into 144 orthogroups, of which 97 orthogroups with ≤5 candidate NLRs were visually inspected and corrected for gene prediction errors, and 47 orthogroups with >5 candidate NLRs were subjected to additional annotation steps. We generated HMM profiles from NB domains for all 47 orthogroups. These profiles were used to detect genes with NB domains initially missed during domain prediction. Among genes with an NB annotation, we only kept genes longer than 1kb and we aligned the remaining 3,758 NLR candidates against the 84 genome assemblies using Exonerate to (i) correct truncated NLRs, (ii) identify NLRs missed by gene prediction, and (iii) identify NLRs with non-canonical structures, either due to the presence of non-canonical domains or the lack of any domain at the C-terminus, N-terminus, or both. The final set of NLRs included 4769 NLR sequences: 4613 within the 82 *Sordariales* genomes (S2 Table) and 153 within the two outgroups. Orthology analysis of the 4613 candidate NLR sequences identified 485 NLR orthogroups: 91 in *Chaetomiaceae*, 229 in *Lasiosphaeriaceae*, 119 in *Podosporaceae*, and 46 in *Sordariaceae* (S3 and S4 Tables). The majority of NLRs (3,858 – 84%) clustered in 191 large orthogroups (39%) comprising six or more NLRs. The remaining NLRs (688 – 15%) were grouped into 227 (47%) orthogroups containing two to five NLRs, and 67 NLRs (1%) were singletons. In most orthogroups, NLRs were unevenly distributed among the different species, with only 14 orthogroups (3%) including sequences from all species of the corresponding *Sordariales* family: 10 in *Podosporaceae*, four in *Lasiosphaeriaceae*, one in *Sordariaceae*, and none in *Chaetomiaceae*. Most of the large orthogroups (153 – 80%) and many of the small orthogroups (64 – 28%) were species-specific and thus included paralogous sequences.

The average number of NLRs per *Sordariales* genome was 56 (s.d.: 42.8), and ranged from 10 NLRs in *Crassicarpon thermophilum* and *Trichocladium antarcticum*, to 162 NLRs in *Lasiosphaeria ovina* (Fig 2D and S1 Table). At the family level, the average number of NLRs was 60 in *Chaetomiaceae* (s.d.: 34.7; coefficient of variation [CV]: 0.58 +/- 0.19), 122 in *Lasiosphaeriaceae* (s.d.: 25.8; CV: 0.21 +/- 0.11), 30 in *Sordariaceae* (s.d.: 26.4; CV: 0.89 +/- 0.17), and 71 in *Podosporaceae* (s.d.: 21.6; CV: 0.31 +/- 0.37). A Kruskal-Wallis test indicated a significant family effect on the number of NLRs,

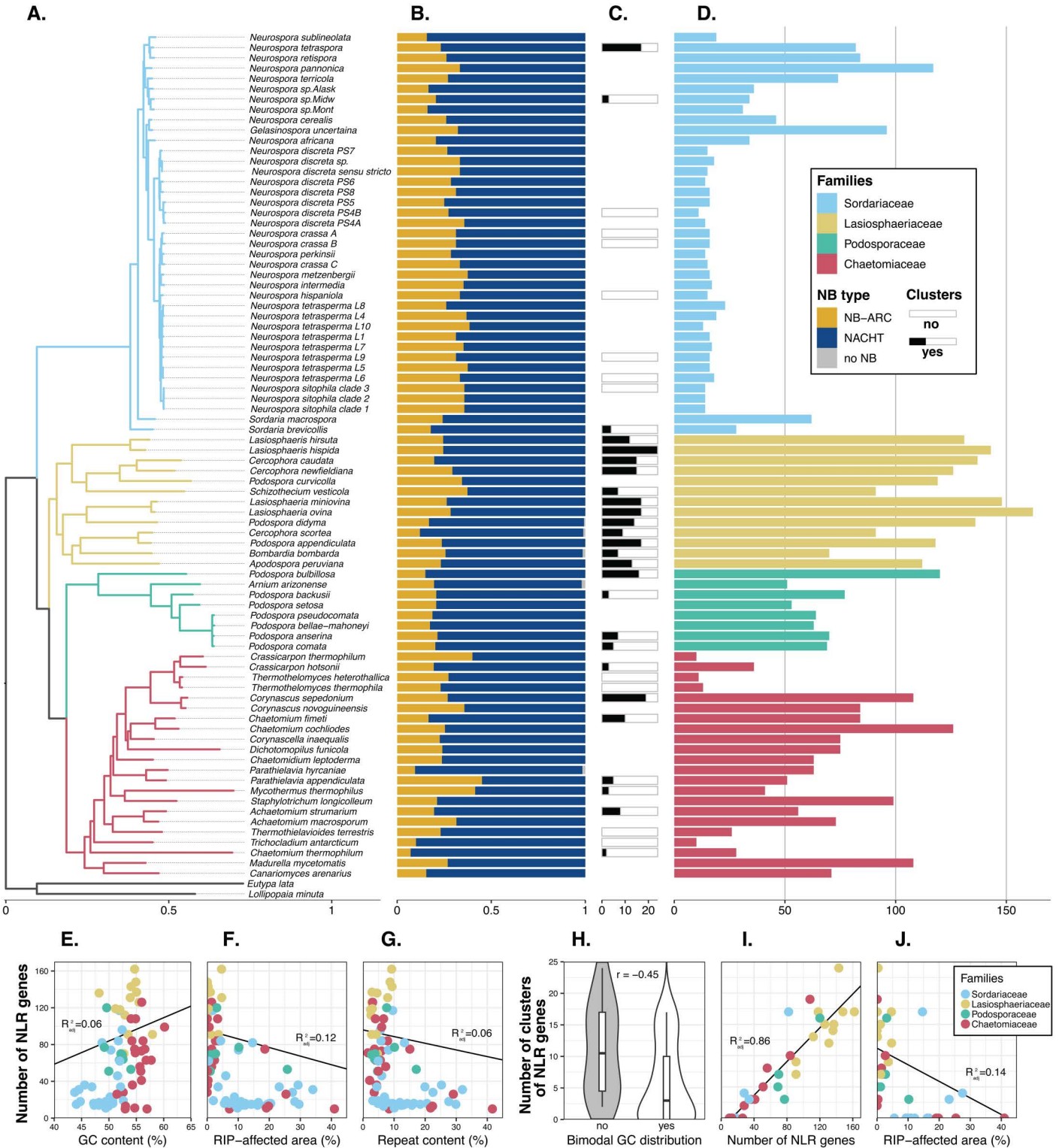

**Fig 2. NLR richness across *Sordariales*.** (A) *Sordariales* species tree built with ᴀꜱᴛʀᴀʟ-III and 2367 gene trees; branch length was estimated using RAxML-ng and 1030 single-copy orthologs present in all taxa. In panels A, D-G, I, and J, colors represent Sordariales families, as defined by Wang *et al.* [49]. (B) Frequency of the different types of NB domains in individual NLRomes. (C) Number of NLR clusters identified in genomes with an N50 greater

than 1Mb. (D) Number of NLR genes per genome. (E-F-G) Relationships between the number of NLR genes and the GC content, the RIP affected area, and the repeat content. (H-I-J) Relationships between the number of NLR clusters and the bimodal GC distribution, the number of NLR genes, and the RIP-affected area. For continuous parameters, a linear regression is presented, and the adjusted R² is reported. For the bimodal GC distribution, a correlation index r was calculated using a threshold model.

and post-hoc Wilcoxon-Mann-Whitney tests revealed statistically significant differences between all pairs of families, except *Chaetomiaceae* and *Podosporaceae*. At the family scale, we observed extensive variation in NLR content within *Sordariaceae*, with fire-associated *Neurospora* taxa presenting smaller repertoires than their soil- or dung-associated relatives. Using phylogenetic generalized least squares to account for knowledge of phylogenetic relationships, we found no significant association between the number of NLRs and assembly quality statistics (N50, L50, number of contigs, total length, or Busco score; S5 Table), indicating that the observed variation in NLR content is primarily biological in origin.

### No strong relationship between mechanisms regulating genomic repeat elements and NLR content

The variation in NLR content between Sordariales families suggests historical duplication events in certain lineages despite defense mechanisms against genomic repeat elements. To examine the relationship between the number of NLRs and the stringency of mechanisms involved in regulating genomic repeat elements, such as repeat-induced point mutations (RIP) [38], we used phylogenetic generalized least squares [50] and a threshold model [51]. RIP is a mechanism of fungal genome defense that induces C-to-T mutations in repetitive DNA to prevent their expression. The number of NLRs was positively significantly correlated with the GC content (adjusted $R^2$: 0.07; Fig 2E) and negatively significantly correlated with the repeat content (adjusted $R^2$: 0.06; Fig 2G) and the percentage of RIP-affected regions (adjusted $R^2$: 0.12; Fig 2F), but correlations were relatively weak (S5 Table). Transposon invasion followed by RIP can lead to genomic compartmentalization of AT-rich regions and a bimodal distribution of GC content [52]. We found that differences in the number of NLRs between genomes with and without bimodal distributions of GC content were not statistically significant. Together, these analyses indicated that mechanisms of defense against genomic repeat elements had no strong impact on the number of NLRs.

### NLRs form clusters in NLR-rich species

The lack of a clear correlation between NLR content and RIP stringency should not be seen as evidence that NLRs evade defense mechanisms, but rather as a sign that they have diversified through processes other than tandem gene duplication. To better understand how NLR repertoires diversified in the *Sordariales*, we studied the presence of NLR clusters. In plants, NLRs often form clusters, which may represent a means for generating new functional diversity through unequal crossovers (leading to gene loss and the formation of new genes), as well as gene conversion events between partially orthologous genes [25,53,54]. These changes in sequence and genomic organization may be facilitated by the repetitive structure of the C-terminal region of NLRs.

Before determining whether the NLRs form clusters, we had to define what a cluster is in the *Sordariales*. Plant NLR clusters are usually defined as sets of NLRs separated by less than 200 kb and by a limited number of other genes, usually up to eight [35]. However, this definition may not be suitable for fungal genomes, which are generally more gene-dense than plant genomes. We therefore sought to determine the optimal genomic size and number of NLRs to define clusters in *Sordariales*. We focused on the 37 *Sordariales* genomes with an N50 higher than 1Mb, in which 2349 NLRs were identified. We then compared the number of clustered genes for NLRs and 1000 sets of randomly-selected genes while varying the maximum distance and the number of non-NLR genes between NLRs of the same cluster. For each genome, the size of each of the 1000 sets of randomly-selected genes was equal to the number of NLRs. We observed that for genomes in which NLR genes formed more clusters than randomly-selected genes, the difference in the number

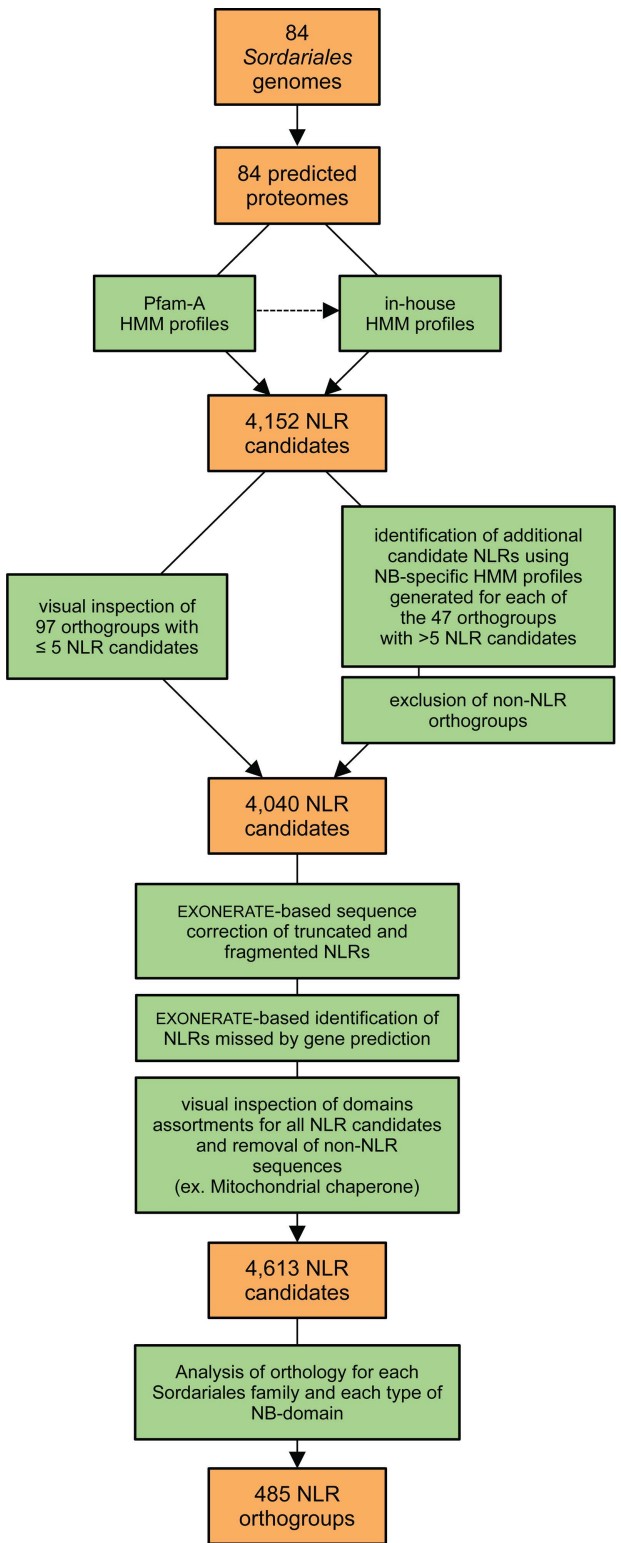

**Fig 3. Bioinformatic identification of NLRs in 84 *Sordariales* genomes, with counts of the number of candidate genes and orthogroups identified.**

of clusters was maximized for cluster length varying between 20–45 kb and with a maximum of eight non-NLR genes between two NLRs. Based on these simulations, we defined a *Sordariales* NLR cluster as a group of two or more NLRs, with two consecutive NLRs being separated by less than 40 kb and less than eight non-NLR genes.

Using these criteria, we identified that 602 (26%) of the 2349 NLRs formed 272 clusters in 26 genomes (Figs 2C and 4 and S1 and S6 Tables). No NLR clusters were identified in the remaining eleven genomes, corresponding to the seven conidiating species of *Neurospora* (*N. crassa* clade A, *N. crassa* clade B, *N. hispaniola*, *N. tetrasperma* L9, *N. tetrasperma* L6, *N. sitophila* clade 3, *N. discreta* PS4B) and four *Chaetomiaceae* species (*Thermothelomyces heterothallica*, *Thermothielavioides terrestris*, *Thermothelomyces thermophila*, *T. antarcticum*). We then examined the cluster size, the variability of domain assortments within clusters, and estimates of the stringency of defense against genomic repeats, to gain insight into the mechanisms underlying cluster formation. Among the 272 clusters, 164 consisted of NLRs sharing the same type of NB domain, and only 16 clusters consisted of paralogous NLR copies. The number of NLRs per cluster was relatively limited, with 225 clusters of two NLRs, and 47 clusters of three to five NLRs. The numbers of NLR genes and NLR clusters were highly and significantly positively correlated (adjusted $R^2$: 0.86; Fig 2I). No strong associations existed between the number of NLR clusters and assembly quality. Among assembly quality statistics (N50, L50, number of contigs, total length, or Busco score), only the Busco score showed a positive, significant but relatively weak correlation with the number of NLR clusters (adjusted $R^2$: 0.12; S5 Table). GC and repeat content did not correlate with the number of NLR clusters. Both the RIP bimodal distribution (mean correlation r: -0.45; 95% confidence interval not overlapping zero; Fig 2H) and the

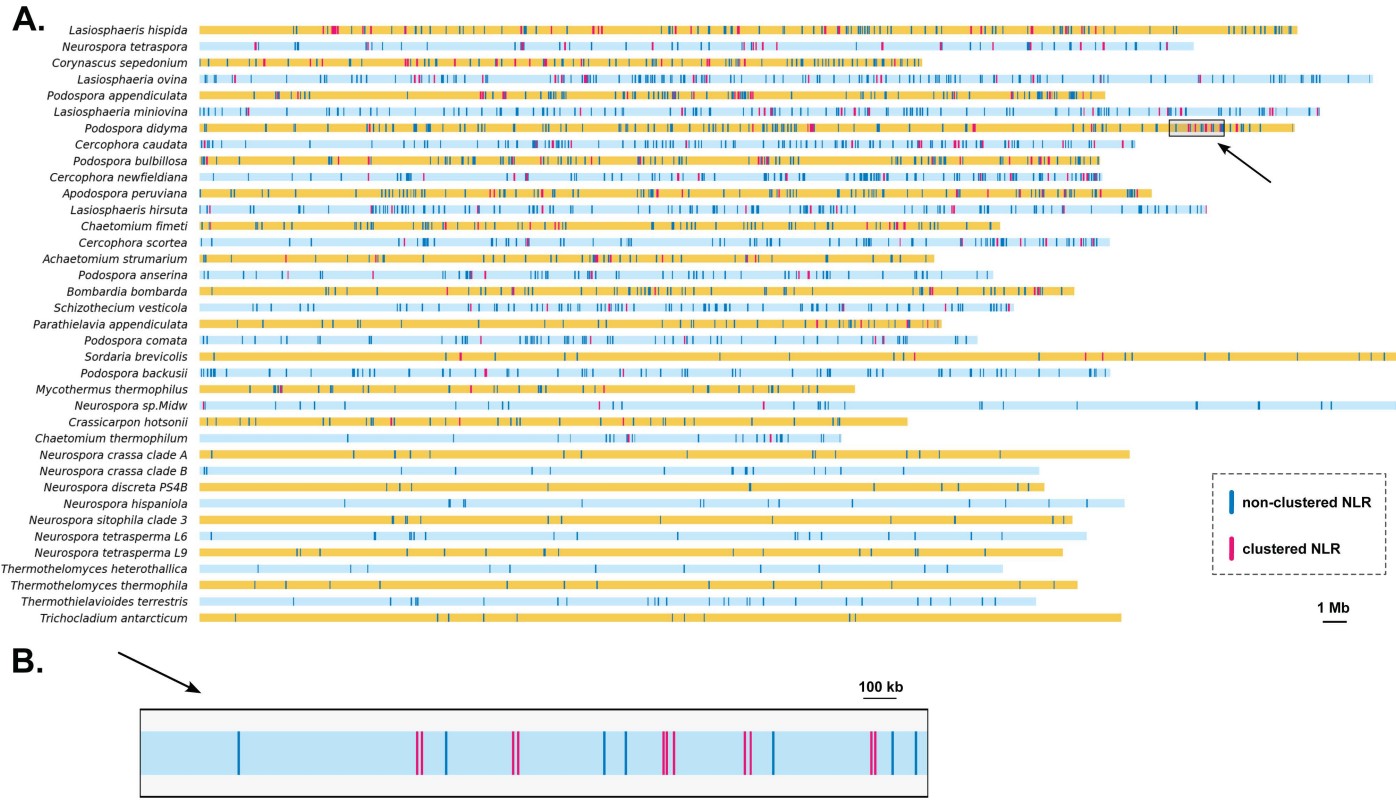

**Fig 4. Distribution of NLRs along *Sordariales* genome.** (A) Distribution of non-clustered NLRs (blue) and clustered NLRs (red) along the 26 genomes with an N50 larger than 1Mb. All scaffolds have been concatenated to simplify the representation. Genomes are organized from top to bottom in decreasing number of clustered NLRs. (B) Distribution of the seven non-clustered NLRs (blue) and the 11 clustered NLRs (red) along the scaffold 10 of *Podospora didyma*.

percentage of RIP-affected regions (adjusted $R^2$: 0.14; Fig 2J and S5 Table) showed a negative significant correlation with the number of NLR clusters. These results, particularly the correlation between the number of NLRs and the number of clusters, suggest that organizing into clusters may enable NLR repertoires to diversify. RIP prevents tandem gene duplications, but processes such as unequal crossing over may allow the formation of clusters without strict duplications.

Another important feature of plant NLRs whose existence we sought to examine in Sordariales was their frequent organization as head-to-head pairs. Paired NLRs in plants follow a "sensor-helper" model in which the sensor NLR binds the exogenous ligand while the helper NLR triggers the immune response. The NLR sensor often includes an integrated decoy [55]. Among the 225 clusters of two NLRs, 13 were organized as head-to-head NLRs, identified in 11 genomes (S7 Table). Most head-to-head NLRs (19/26) had a complete tripartite structure, and only one paired NLR contained a non-canonical domain (Lipocalin_5; S7 Table). These results indicate that head-to-head NLRs are rare in *Sordariales* and that *Sordariales* NLRs have not converged evolutionarily towards the same type of head-to-head organization as plant NLRs.

### *Sordariales* display two types of NACHT NLR proteins

As a minority of 55 NLRs presented an NB domain (Fig 2B) that could not be identified as NACHT or NB-ARC, we set out to clarify their identity. Out of 4613 *Sordariales* NLRs, 3350 (73%) presented a NACHT domain, 1141 (25%) an NB-ARC, and 55 NLRs were identified as both NACHT and NB-ARC, with close E-values in analyses with HMMER. In the NLR-specific orthology analysis, all 55 NLRs clustered with an additional seven NACHT NLRs, one NB-ARC NLR, and seven NLRs without identified NB domain, forming four family-specific orthogroups. To clarify the identity of the NB domain present in these 70 NLRs, we built an alignment and examined the Walker B motif that is expected to distinguish NB-ARC and NACHT domains [23]. We observed that the first acidic residue (aspartic acid) of the Walker B was not directly followed by another acidic residue, as expected for the NB-ARC, but by a glycine. We also found that another acidic residue (aspartic acid) was present three positions downstream of the first one. These features fitted the typical NACHT consensus motif hhhhD[GAS]hDE described in Leipe *et al.* [23].

To ascertain why our HMM profiles failed to identify a minority of NLRs with a typical NACHT Walker B as NACHT domains, we further examined their similarity with other NACHT and NB-ARC domains, using four sequences of these NACHT domains, 325 sequences of other NACHT domains, and 85 sequences of NB-ARC domains. All sequences represented different orthogroups. We split NB sequences into two fragments (from the N-terminal end to the start of the Walker B motif, and from the start of the Walker B to the C-terminal end; Fig 5A). For each fragment, we computed distances between sequences using alignment-free methods of sequence comparison [56,57], and visualized distance matrices using principal coordinates analysis (PCoA). In PCoAs based on the first and second fragment (Figs 5E-H and S2), NB-ARC and *bona fide* NACHT fragments were clearly distinguished along the first coordinate. In the PCoA based on the distance between C-terminal sequences (*i.e.*, second fragment; Figs 5F, 5H and S2), the four unclassified NB sequences clustered with other NACHT sequences, apart from the NB-ARC sequences. In the PCoA based on the distance between N-terminal sequences (*i.e.,* first fragment; Figs 5E, 5G and S2), the four unclassified NB sequences were intermediate between NACHT and NB-ARC sequences. Thus, these distance-based analyses indicate that the four unclassified NB sequences, while matching the NACHT definition based on the Walker B motif, present dissimilarities with the other NACHT sequences identified in *Sordariales* NLRs.

To identify the genomic features that differentiate the four unclassified NB sequences from the other NACHT sequences, we examined the HElical THird domain of STAND proteins (HETHS), also named winged helix domain or ARC2 domain, which is a helical bundle located at the C-terminus of the core P-loop domain of STAND proteins (Fig 5A). Leipe *et al.* [23] proposed a subdivision of NACHT genes into two families: the NAIP-like and TLP1-like families, partly based on the HETHS domain. The NAIP-like family is defined as having a consensus motif FhHxxhQE[YF]hxA in the HETHS domain (note that we corrected a typo in the motif reported in the original publication). Unlike other NACHT

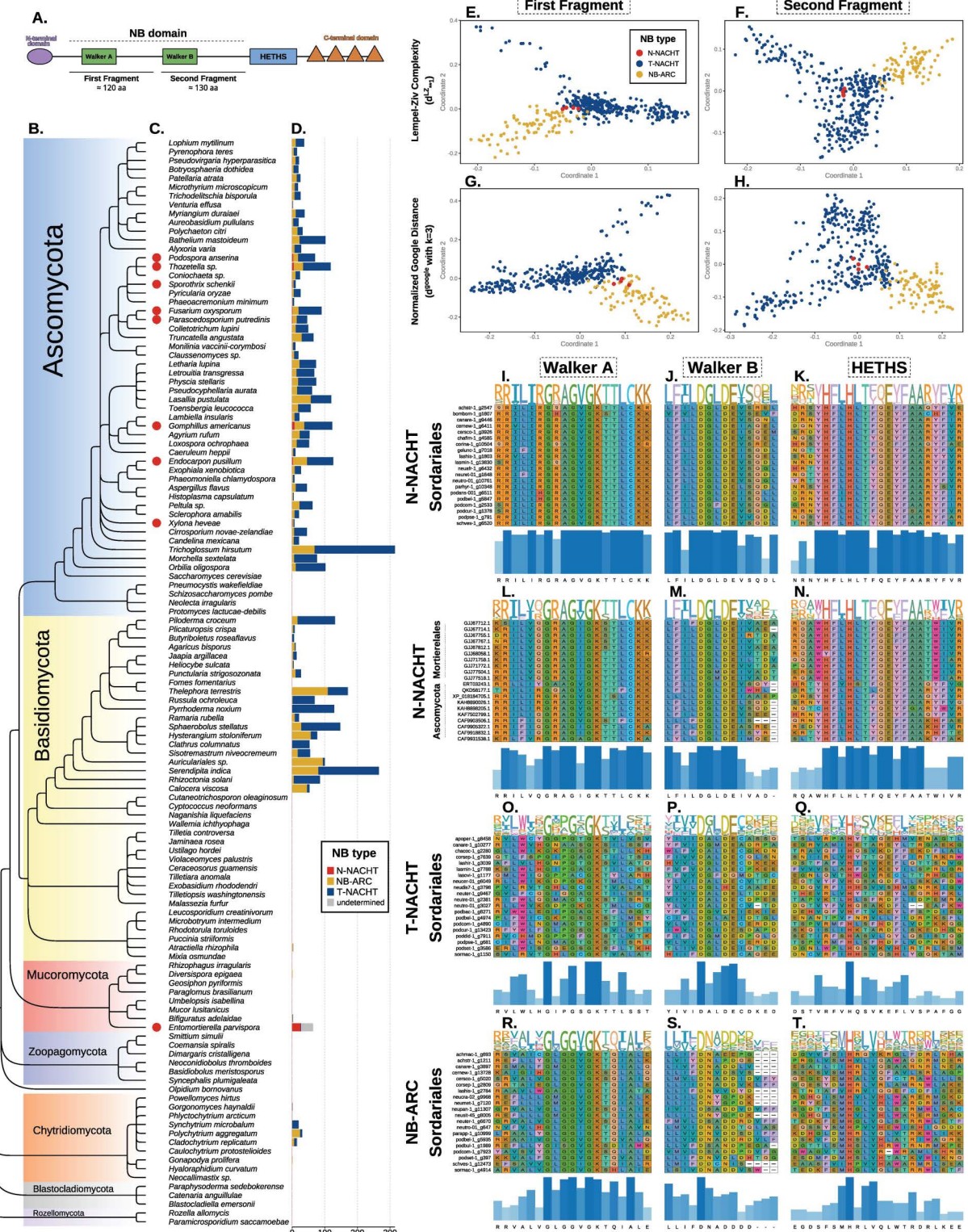

**Fig 5. Two types of fungal NACHT NLRs: the N-NACHT and the T-NACHT.** (A) Domain structure of an NLR, showing the two fragments considered in remaining panels, and the relative position of the HETHS. (B) Cladogram of 122 fungal species representative of the whole fungal kingdom adapted from James *et al.* [58]. (C) Red dots indicate genomes in which N-NACHT NLRs were identified. (D) Absolute numbers of N-NACHT, NB-ARC, T-NACHT

NLRs, and NLRs with an undetermined NB domain. NLRs were defined as genes containing an NB and a canonical C-terminal domain. (E-H) Principal coordinate analyses of matrices of alignment-free genetic distances of the first and second NB fragments of NACHT and NB-ARC NLRs as defined in (A). Normalized Google Distance (G-H) is based on 3-mers, while the Lempel-Ziv Complexity distance (E-F) relies on information theory. (I-T) Alignment of the Walker A, Walker B, and HETHS for 20 N-NACHT NLRs of *Sordariales*, 20 N-NACHT NLRs of non-*Sordariales* fungi (10 from *Ascomycota* and 10 from *Entomortierella parvispora*), 20 T-NACHT NLRs of *Sordariales*, and 20 NB-ARC NLRs of *Sordariales*.

sequences, our four representatives of divergent NACHT sequences presented the consensus FLHLT[FY]QEYFAA at the HETHS domain (Fig 5K), which placed them in the NAIP-like family of NACHT domains, while other NACHT sequences represented the TLP1-like family (Fig 5Q). In the following, the two domains are referred to as N-NACHT for the NAIP-like form (which is rare) and T-NACHT for the TLP1-like family (which is more frequent).

To re-evaluate the distribution of NB domains considering two types of NACHT domains, we constructed HMM profiles based on the alignments of N-NACHT on the one hand, and T-NACHT on the other hand. Among the 4,613 *Sordariales* NLRs, T-NACHT was the most common NB domain (3388 NLRs, 73.4%), followed by NB-ARC (1150 NLRs, 24.9%), and N-NACHT (70 NLRs, 1.5%). The NB domain could not be identified for five NLRs (0.1%), which do not cluster with any NLR with annotated NB in orthology analyses. These five genes were still considered NLRs because they passed our cutoff on identity levels in similarity analyses with EXONERATE. All genomes harbored both T-NACHT and NB-ARC NLRs, but only 33 of the 82 genomes had N-NACHT NLRs, with one to five N-NACHT NLR genes per genome. Together, these analyses show that using the diagnostic motifs of the different types of NB domains allows the building of HMM profiles that detect NB domains with increased sensitivity.

## Origin of N-NACHT NLRs in Sordariales

To investigate the origin of N-NACHT NLRs in *Sordariales*, we characterized their distribution throughout the Fungal Kingdom. We assembled a dataset of 122 proteomes representative of the whole Fungal Kingdom (Fig 5B and S8 Table). Using both Pfam-A and HMM profiles, we predicted NLRs within these proteomes, which we defined as genes presenting both an NB domain and canonical C-terminal repeats. We identified a total of 3993 NLRs in 77 of the proteomes (Fig 5D and S9 Table), with NLR numbers ranging from one in species of *Chytridiomycota* and *Mucoromycota* to 315 in *Trichoglossum hirsutum* (*Ascomycota*: *Helotiales*). NLRs were missing in *Saccharomyces* and *Schizosaccharomycota* species, in the earliest branching lineages of *Basidiomycota*, and in some of the early branching lineages of fungi (*Zoopagomycota*, *Blastocladiomycota*, and *Rozellomycota*). Among the 3993 fungal NLRs, we identified 2722 T-NACHT NLRs, 1180 NB-ARC NLRs, 44 N-NACHT NLRs, and 47 NLRs with an undetermined NB (Fig 5D and S9 Table). N-NACHT NLRs were identified only in seven species of *Ascomycota* and one species of *Mucoromycota* (*Entomortierella parvispora, Mortierellales*) (Fig 5C and S9 Table). The number of N-NACHT NLRs ranged from one to seven in the seven *Ascomycota* species, but reached 26 in *Entomortierella parvispora*. We conclude from these analyzes that N-NACHTs are not an innovation specific to *Sordariales*, but that they have been acquired or lost several times during fungal evolution.

## NLR domain assortments in *Sordariales* represent a subset of the larger *Dikarya*

At the scale of *Dikarya*, NLRs exhibit high diversity of N-terminal and C-terminal domains, and a wide variety of combinatorial domain assortments [2,16]. We aimed to investigate whether this pattern persisted on a smaller evolutionary scale, within *Sordariales*, or if previous measurements of NLR repertoire variability at the *Dikarya* scale had been underestimated due to using a limited number of genomes per phylum. We annotated the three types of NB domain (N-NACHT, T-NACHT and NB-ARC), N-terminal and C-terminal domains (Fig 6A) in *Sordariales* using Pfam-A or Sordariales-specific HMM profiles. We next assigned C-terminal repeats to four categories of canonical (*i.e.*, identified as main components of NLR architectures) NLR repeats (TPR, Ankyrin repeats, WD40 repeats, and HEAT) for 3104 NLRs (67%), and N-terminal repeats to eight canonical N-terminal domains (Goodbye-like, HeLo-like, HET-like, Patatin, PNP_UDP_1, RelA_SpoT,

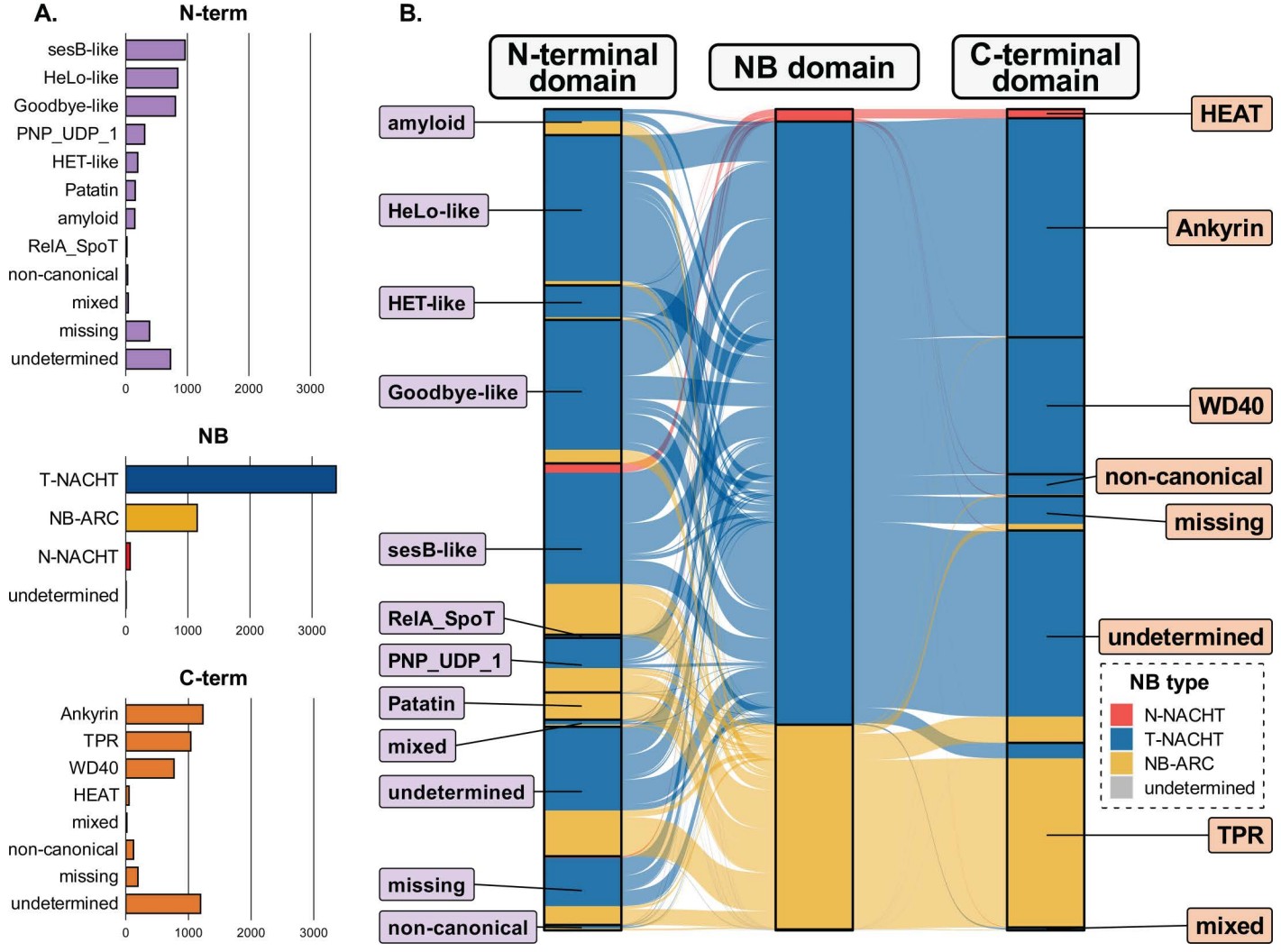

**Fig 6. Architectural variability of NLRs in *Sordariales*.** (A) Distribution of NLRs based on their canonical N-terminal, NB, and C-terminal annotations. (B) Associations between canonical N-terminal, NB, and C-terminal domains. Category 'undetermined' represents NLRs without any domain predicted. Category 'missing' represents truncated NLRs (<20 aa in N-terminus and <50 aa in C-terminus). Category 'mixed' represents NLRs with several canonical domains. Category 'non-canonical' represents NLRs without a canonical domain but with a non-canonical domain.

sesB-like, or amyloid) for 3324 NLRs (72%) (Fig 6A). The most frequent N-terminal domain across *Sordariales* were sesB-like, HeLo-like, Goodbye-like, the most frequent NB domain was T-NACHT, and the most frequent C-terminal repeats were Ankyrin, TPR, and WD40 (Fig 6A). The most frequent NB/C-terminal domain associations were NB-ARC/TPR for NB-ARC NLRs, T-NACHT/Ankyrin or T-NACHT/WD40 for T-NACHT NLRs, and N-NACHT/HEAT for N-NACHT NLRs (Fig 6B). These most frequent NB/C-terminal domain assortments represented 2990 (96%) of the 3104 NLRs exhibiting a canonical C-terminal domain. Among the NLRs with a canonical N-terminal domain, most NLRs with a HeLo-like domain (820 – 97%), a HET-like domain (179 – 92%), a Goodbye-like domain (729 – 90%), or a RelA_SpoT domain (14 – 82%) were T-NACHT NLRs. NLRs with a Patatin domain were almost exclusively of the NB-ARC type (146 – 95%). PNP_UDP_1 and amyloid domains were associated with T-NACHT (170 – 54% and 68–44%, respectively) or NB-ARC (145 – 46% and 87–56%, respectively). The sesB-like domain was the only canonical N-terminal domain that was associated

with all three types of NB domains: T-NACHT (624 – 65%), NB-ARC (287 – 30%), and N-NACHT (53 – 5%) (Fig 6B), and the only canonical N-terminal domain identified in N-NACHT NLRs.

NLRs with similar architectures were generally grouped into the same orthogroups. Most large orthogroups (≥6 NLRs) included a single type of C-terminal domain (166 – 87%) or a single type of N-terminal domain (120 – 63%). Furthermore, for 8% and 17% of large orthogroups, minor C-terminal or N-terminal types, respectively, were represented in only one or two sequences. The NLR architectures in the largest orthogroups reflected the most frequent NB/C-terminal and NB/N-terminal domain associations. For instance, the largest NB-ARC orthogroups included NLRs with TPR repeats, while the largest T-NACHT orthogroups included NLRs with either Ankyrin or WD40 repeats.

Only a subset of all possible domain combinations was observed in *Sordariales*. Out of the total possible 24 (8 × 3) N-terminal/NB and 12 (3 × 4) NB/C-terminal combinations, only 17 and 7 were observed, respectively. Only 33 of the 96 tripartite domain architectures were identified (S10 Table). Hence, not all tripartite combinations were observed in *Sordariales*, but the realized proportion of the 96 possible tripartite domain assortments was only marginally lower in *Sordariales* (39.3%) than previously reported in *Dikarya* (44.4%) [2,16].

### *Sordariales* NLRs encompass a wide variety of non-canonical domains

In addition to canonical domains, additional non-canonical domains can be found in fungal NLRs. In plants NLRs, non-canonical domains are generally rare, but they may play a crucial role in the immune response by acting as integrated decoys, *i.e.,* as domains targeted by pathogens that were duplicated and fused to NLRs to act as baits to trigger the defense response [59]. We examined the presence and putative function of non-canonical domains in the *Sordariales* to determine if they could correspond to *bona fide* integrated decoys or if, on the contrary, they are domains that should be added to the list of canonical domains.

We identified 185 non-canonical domains in 378 *Sordariales* NLRs (Fig 7A and S11 Table). Non-canonical domains were identified either at the N-terminal or C-terminal regions, either on their own (indicated in Fig 6B), connected with a canonical domain, or connected with another non-canonical domain. Most of these non-canonical domains (170–92%) were found solely in one to three NLRs. The 185 domains were identified in 327 T-NACHT NLRs (10%), 50 NB-ARC NLRs (4%), and one N-NACHT NLR.

The three most abundant types of non-canonical domains were binding domains of types zinc-finger, SPRY and PKinase (Fig 7A). We identified zinc-finger domains in 57 NLRs, in most cases in the C-terminal position (52 NLRs), and in five cases as the only N-terminal domain. Thirty-four NLRs contained C2H2 zinc-finger type, while 23 NLRs contained treble clef zinc-finger type (17 ZZ, three C1, two FYVE and one TRASH) [60]. Only 16 zinc-finger domains, all of treble clef type, were located next to canonical C-terminal repeats (Fig 7B). A SPRY domain was found in 46 NLRs, always found in the C-terminal position and always associated with Ankyrin repeats (Fig 7B). These analyses indicate that the most frequent non-canonical domains were binding domains, and consistent with their binding activity, they were predominantly located in the C-terminal position.

PKinase domains were the third most common non-canonical domains (12 NLRs; Fig 7A), and nine Pkinase domains represented good candidates for being integrated decoys. PKinase domains were found at the C-terminus of nine T-NACHT NLRs in two orthogroups exclusive to *Chaetomiaceae* and *Podosporaceae*. Aside from the PKinase domain, all nine NLRs had a complete canonical tripartite architecture (Fig 7C). We built a gene tree for all NB domain sequences from both orthogroups (S3 Fig). The genealogy of the NB domain revealed that all nine PKinase-including NLRs were monophyletic and thus that all nine PKinase domains may have originated from a single integration event. To test the hypothesis of a single integration event, we compared PKinase sequences of the nine NLRs with all *Sordariales* proteomes. We observed high similarity between the NLR PKinases and other PKinase domains in single-copy core genes identified as CHK-2 proteins. We calculated BLOSUM62 distance index between all pairs of PKinase domains, both from NLRs and CHK-2 proteins (S12 Table). BLOSUM62 distance was low among CHK-2 sequences (between 0 and 0.29)

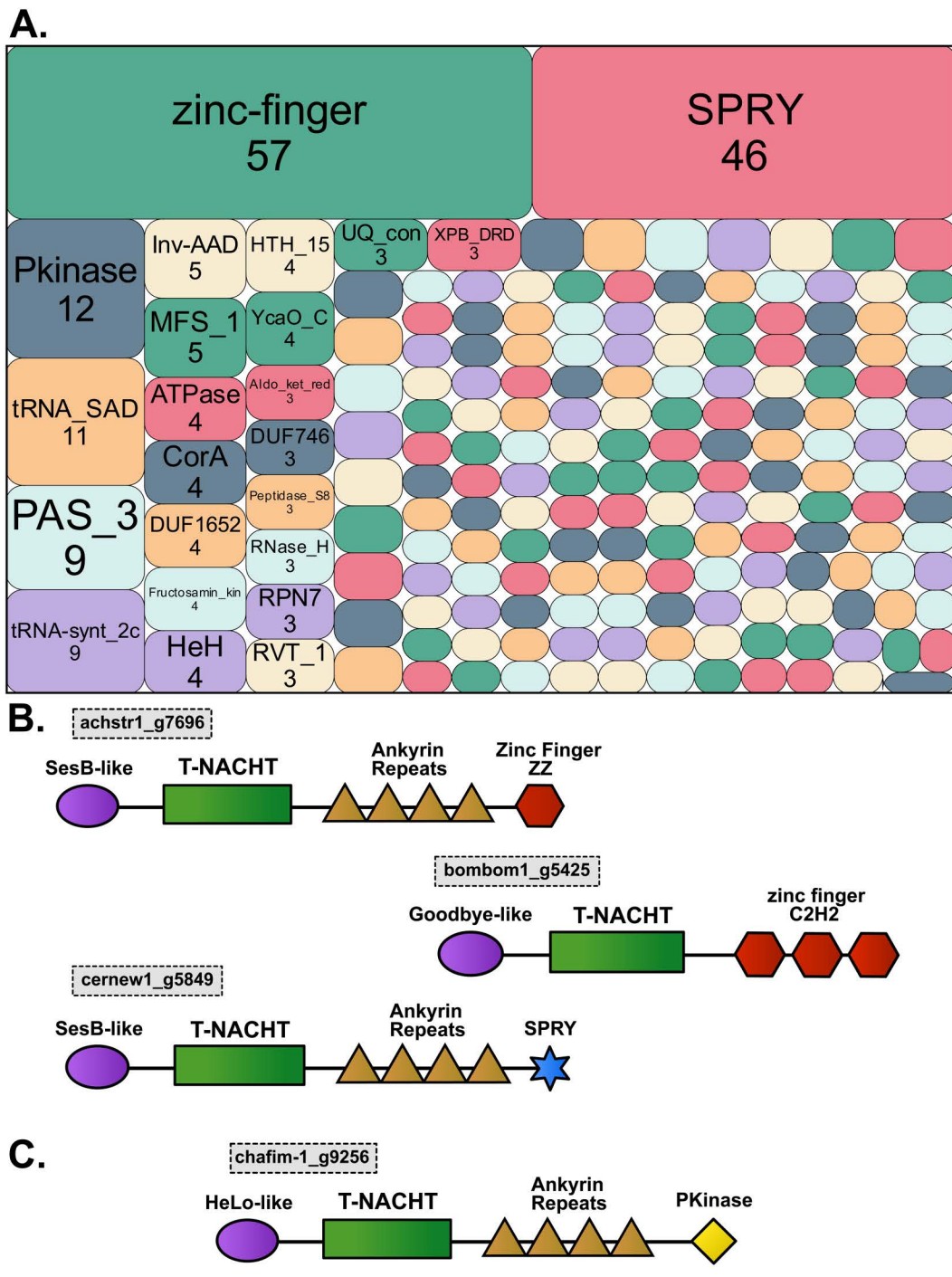

**Fig 7. Fungal NLRs include non-canonical domains.** (A) Number of NLRs with non-canonical domain annotations, with surface area proportional to the number of NLRs. Names and numbers of NLRs are specified only for the most frequent domains. (B) Examples of three NLR architectures that include a zinc finger domain or a SPRY domain in C-terminal position. The hexagons repeated in tandem in NLR bombom-1_g5425 reflect that the zinc finger motif was repeated three times in the C-terminal position. In the two other NLRs, the zinc finger and SPRY motifs were found only once. These three NLRs are characteristic of architectures containing zinc finger and SPRY domains. (C) Architecture of NLR chafim-1_g9256 of *Chaetomium fimeti*. This NLR includes a PKinase domain in C-terminal position, which could act as a sensor to detect pathogen-derived effectors.

and NLR sequences (between 0.04 and 0.46), as well as between CHK-2 and NLR sequences (between 0.32 and 0.45). The similarity between PKinase sequences in NLRs and CHK-2 proteins supported the hypothesis that their PKinase domain was integrated from a CHK-2 gene and maintained as a putative integrated decoy.

## Discussion

Fungal NLRs have previously been shown to be characterized by a great variability of domain organizations [2,4], but the identity of the nucleotide-binding domains, the genomic localization, and the factors associated with variation in the composition of repertoires of fungal NLRs have remained largely unknown. Herein, we conducted a systematic exploration of NLR diversity in genome data from the ascomycete order *Sordariales* to gain a better understanding of the variability of fungal NLR repertoires and the factors underlying variability. We found that the number of NLR genes was highly variable and that NLRs were organized as clusters in NLR-rich species. By improving the annotation of the NB domain, we also showed that the NACHT domain of fungi comes in two forms, N-NACHT and T-NACHT.

### Extensive variation in NLR repertoire size among *Sordariales* species

Variation in the number of NLRs in *Sordariales* was extensive. The variation in the number of NLRs could be as much as sixteen times amongst different species of the order, and between two to thirteen times within a family. The variation in repertoire size was also shown at the broader *Dikarya* scale, ranging from zero NLRs in *Saccharomyces sp.* to 602 NLRs in *Fibularhizoctonia sp.* [2,4]. The variation observed in *Sordariaceae*, the family of *Neurospora* relatives, however, allows us to gain insight into the ultimate (*i.e.*, eco-evolutionary) factors underlying NLR diversification. Indeed, dung- or soil-associated *Neurospora* species harbored, on average, over three times as many NLRs as conidiating, wild fire-associated species of *Neurospora* (*N. crassa*, *N. discreta*, *N. perkinsii*, *N. metzenbergii*, *N. intermedia*, *N. hispaniola*, *N. tetrasperma*, and *N. sitophila*). Differences in repertoire size in *Neurospora* and hypothetically in the broader *Sordariales* could be explained by variation in the diversity of mycoparasites on and in plants or soil, assuming NLRs are significant players in fungal innate immunity. To investigate this hypothesis further, a better understanding of the life cycle of *Sordariales* is required.

### NLRs can be organized in clusters in NLR-rich species

Our data also allowed us to gain insight into the proximal (*i.e.*, genomic) factors underlying variation in NLR repertoire size. We discovered that the relationship between NLR content and the stringency of defense mechanisms against genomic repeat elements (*e.g.*, RIP) was weak, indicating that NLRs have diversified by means other than tandem gene duplication. The organization of Sordariales NLRs into clusters, in particular, could be one of the proximal factors explaining the size of the repertoires. Having proposed a definition of NLR clusters that consider the higher gene density of fungi compared to plants, we searched for such clusters in the 37 *Sordariales* genomes with the longest assembled scaffolds. Most *Sordariales* genomes presented clusters of NLRs, and the numbers of NLR clusters and NLR genes were highly positively correlated, while the number of NLR clusters was weakly correlated with the strength of RIP. In species that exhibited NLR clusters, approximately 25% (maximum 46%) of NLRs were clustered, although clusters were relatively small, with less than five NLR genes. This trend is in contrast to what is observed in plants where often more than 50% of NLRs are clustered, and these clusters can contain over 10 NLR genes [35,61]. Regardless, clusters could contribute to diversification by promoting unequal crossing over and gene conversion among partially homologous genes, favored by the repetitive nature of the C-terminus of NLRs.

### Improved domain annotation in *Sordariales* uncovers three types of nucleotide-binding domains

Using a combination of Pfam-A and *Sordariales*-specific HMM profiles, we were able to substantially improve the annotation of nucleotide-binding domains compared to previous studies [2,4], with only 0.4% of NB domains remaining

undetermined. By examining the Helical Third section of the nucleotide-binding domains [23], we were able to demonstrate that, similarly to animals, fungi have NACHT domains of both NAIP-like and TLP1-like types, which are referred to as N- and T-NACHT, respectively. Unlike animals, where the rare form of NACHT (N-NACHT) is almost exclusively associated with LRRS, it is exclusively associated with HEAT repeats in fungi. Recently, it was shown that the N-NACHT/HEAT architecture is also present in *Rickettsiales* [17], an order of obligate intracellular proteobacterial endosymbionts of animals, suggesting a possible origin through horizontal transfer. To explain the patchiness of the distribution of this architecture, the hypothesis of a single ancestral horizontal transfer from *Proteobacteria* to *Opisthokonta*, followed by multiple loss events, is more parsimonious than independent transfers in fungi and animals. Further comparative studies with a high level of phylogenetic density should help clarify the origin of the N-NACHT/HEAT architecture.

### Variability of NLR domain assortments in *Sordariales*

The repertoire of domain assortments identified in the *Sordariales* represents a subset of what is observed in *Dikarya* [2,4] but with only a small part of tripartite combinations in *Dikarya* not found in *Sordariales*. None of the major N-terminal, NB, or C-terminal domains detected in the *Dikarya* were absent from the *Sordariales*. Eight types of N-terminal domains were identified in *Sordariales*, and among the previously reported N-terminal domains [2,4], only minor N-terminal domains such as C2 and Peptidase_S8 were rare or absent in *Sordariales*. Identifying the NAIP-like type of NACHT also allowed us to demonstrate that HEAT repeats represent a major type of C-terminal domain in fungal NLRs, not merely a sub-type of TPR repeats. These results indicate that previous measurements of NLR repertoire variability at the *Dikarya* scale were not underestimated due to the limited number of species included per sub-phylum. Our findings also indicate that the assortment of domains in *Sordariales* is ancient and has mostly been preserved as inherited from their *Dikarya* ancestor.

### Sordariales NLRs harbor putative integrated decoys

We systematically explored non-canonical domains in *Sordariales* NLRs, which allowed us to identify 185 different non-canonical domains, present in 8% of NLRs. The two most commonly found non-canonical domains were zinc-finger and SPRY, which are domains generally described as involved in binding other proteins, amino acids, or molecules [62–64]. These domains were almost exclusively found in the C-terminal position. Zinc fingers were identified in some plant NLRs, but their function remains unknown so far. Some authors state that they could be integrated decoys [39], while others argue for a nucleic acid-binding property [65] or another undetermined property [66]. In humans, the B30.2 domain, which consists of PRY and SPRY domains, is the binding part of TRIM27 in the NOD2-TRIM27 interaction [67]. In teleost, the NLR-B30.2 subgroup comprises NLR proteins with the B30.2 domain as a constitutive part of their architecture [68,69]. In these NLRs, the B30.2 domain likely functions as the pathogen recognition module and is up-regulated during viral infections [33,68]. Given their binding properties and frequency, we hypothesized that the zinc-finger and SPRY domains may act as sensor domains within *Sordariales* NLRs, independently or combined with canonical domains. Thus, these domains may constitute two additional canonical C-terminal domains of fungal NLRs.

Among the non-canonical domains, the PKinase domain, which is the third most frequent, represents a potential integrated domain. PKinase domain proteins are known targets of microbial effectors in plants or animals [39,70,71]. There are many examples of PKinase proteins guarded by NLRs [72], and PKinase domains are often found as integrated domains in plants NLRs [39,55,73]. In fungal NLRs, the PKinase domain has been considered canonical in previous studies [2,4], despite being rare. In *Sordariales*, we identified a PKinase domain in 12 NLRs only. In nine of the 12 NLRs, the PKinase domain was in C-terminal position. These PKinase sequences showed high similarity with a PKinase domain found in a non-NLR single-copy core orthogroup, which includes the PRD-4 gene in *N. crassa*. PRD-4 is a serine/threonine kinase protein, the ortholog of human checkpoint kinase 2 (CHK2) [74]. CHK2/PRD-4 is involved in multiple

pathways in mammals and *N. crassa*, including DNA damage response and circadian cycles [74–77]. In humans, CHK2 can also interact with viral proteins [78]. These observations suggest that the PKinase domain integrated into *Sordariales* NLRs might be a decoy for viruses or microbial effectors targeting CHK2/PRD-4.

## Concluding remarks

Our study aimed to enhance the understanding of NLR domain assortments in *Dikarya* in general, and *Sordariales* in particular. We found that fungi have NACHT domains of both NAIP-like and TLP1-like types, similar to animals. We identified NLR clusters in most taxa, similar to plants, and we provided evidence of an association between the number of NLR and the number of NLR clusters. Thus, our results indicate that fungal NLRs present characteristics shared with fungi's sister group, the animals, and characteristics reflecting evolutionary convergence with plant NLRs. These findings enhance our understanding of the evolutionary history of fungal NLRs and offer new possibilities for comparative immunology.

## Methods

We used various methods to investigate NLR diversity in *Sordariales*. First, we inferred a phylogeny for 82 *Sordariales* isolates and two outgroups. We then identified NLR genes and NLR clusters within *Sordariales* genomes. Next, we used alignment and alignment-free sequence comparisons to show the existence of two types of NACHT domains. Lastly, we performed a functional annotation of NLRs, focusing on canonical and non-canonical domains.

### Dataset

Methods used for genome assembly, gene prediction, and Busco score were previously described in Hensen *et al.* [79]. Our dataset included 84 isolates, with two outgroups, eight *Podosporaceae*, 22 *Chaetomiaceae*, 39 *Sordariaceae*, and 13 *Lasiosphaeriaceae* (sensu Wang *et al.* [49]) (S1 Table). We chose the definition of *Lasiosphaeriaceae* from Wang et al. [49], instead of the one presented by Hensen *et al.* [79], because this former definition is monophyletic and because we had too few (from zero to four) genomes for each of the families represented in *Lasiosphaeriaceae* sensu lato to make meaningful comparisons with the three other families. Each morphological species was represented by one isolate, except *Neurospora discreta*, *N. crassa*, *N. tetrasperma* and *N. sitophila*, for which we included one isolate per subspecific category ("category" is a group [80], lineage [81], clade [82], or phylogenetic species [80,83,84]). Orthology among predicted genes was determined using Orthofinder 2.3.3 using MSA for gene tree inference and Diamond 0.9.22 for similarity searches. Genome assemblies, gene models, HMM profiles, orthogroups and NLR sequences are available from Zenodo (https://doi.org/10.5281/zenodo.14204294).

### Species tree inference

We used two different approaches to infer a species tree of the 84 isolated. The first approach relied on Astral-III (5.7.5). Only single-copy orthogroups with less than 20% species missing (SCO80) were included in the analysis. We used the 2378 SCO80 and not simply the 1030 single-copy core orthogroups (SCCO) because Astral is non-sensitive to gene tree incompleteness (*i.e.,* to missing genes), while reducing the number of gene trees based on missing data may reduce the phylogenetic signal [85]. SCO80 sequences were aligned at the codon level (*i.e.,* keeping sequences in the coding reading frame) with TranslatorX v1.1 [86] using MAFFT as the aligner and default parameters for Gblocks v0.91b [87]. Sequences with more than 60% gaps or missing data were removed [88], sequences were realigned using TranslatorX, and this was repeated until no sequence had more than 60% gaps or missing data. Only alignments over 100 bp were retained, and the final dataset included 2367 SCO80. Gene trees were inferred in RAxML-ng 0.9.0 using the GTR + G model with 50 starting trees (25 maximum parsimony trees and 25 random trees). Branch support was estimated in the same analysis by calculating 1000 bootstrap trees and applying the MRE-based bootstopping test. We detected outliers

using TREESHRINK 1.3.7 with the following parameters: -m per-species -b 20 (a sequence is considered an outlier only if removing it reduces the tree size by more than 20%) -q 0.02 (a sequence is considered an outlier only if its potential tree-size reduction value is among the 2% highest for the individual on the dataset). TREESHRINK detected outliers in 391 gene trees (17%). Outlier sequences were removed from their corresponding SCO80, re-processed with TRANSLATORX and submitted to gene tree inference using RAxML-NG. Genes trees were re-rooted using NW_REROOT (with default parameters) from NEWICK UTILITIES [89]. Low-support branches (bootstrap value <10) were collapsed using NW_ED from NEWICK UTILITIES (option 'i & b<10'). Finally, all 2367 re-rooted and collapsed gene trees were used for species tree inference with ASTRAL-III 5.7.5, with default parameters. Branch lengths for the ASTRAL species tree were estimated using the concatenated sequences, aligned at the codon level, of all 1030 SCCO, using RAxML-NG 0.9.0 with the following options: --evaluate --brlen scaled --opt-model on --opt-branches on.

The second approach inferred a maximum-likelihood genealogy from the concatenation of sequences, aligned at the codon level, of all 1030 SCCO. The concatenated sequence represented 1,530,669 bp. The genealogy was inferred in RAxML-NG 0.9.0 using the GTR + G model with 160 starting trees (80 maximum parsimony trees and 80 random trees). Branch confidence was estimated in the same analysis by calculating 100 bootstrap trees.

### NLR identification

**Generating Sordariales-specific HMM profiles for NB and C-terminal domains.** We used functional annotation of proteomes to identify NLR genes. Three groups of Pfam-A domains were included in NB domain predictions [4]: NACHT, NB-ARC, and AAA. Five groups of Pfam-A domains were included in C-terminal domain predictions [2,4]: Ankyrin repeats, WD40 repeats, TPR, HEAT repeats, and others. All domains are listed in the S13 Table. We searched for these domains using HMMER 3.2.1 for each of the 84 proteomes (hmmsearch -E 1e-6), using the HMM profiles from the Pfam-A database (version 33.1). For each domain, protein sequences of the predicted domains were extracted from the 84 proteomes, and aligned with MAFFT 7.407 (default parameters). Alignments including more than 20 sequences were used to build *Sordariales*-specific HMM profiles using HMMER 3.2.1 (hmmbuild --amino --informat afa), and another round of domain search was carried out (hmmsearch -E 1e-4).

**Predicting a first set of NLR candidates using functional annotation.** A NLR candidate was defined as a gene predicted to encode at least one of the canonical NB domains and one of the canonical C-terminal domains. We identified 4,152 NLR candidates, corresponding to 144 orthogroups. Ninety-seven orthogroups included five NLR candidates or less (162 candidates – 3,9%) and 47 orthogroups contained six NLR candidates or more (3,990 candidates – 96,1%).

**Manual correction of gene predictions for orthogroups with few NLR candidates.** Each of the 97 orthogroups with five NLR candidates or less was aligned using MAFFT 7.407 (default parameters) and inspected visually. Gene predictions (*i.e.,* intron/exon structures) were controlled using RNAseq data (S14 Table) in the Integrative Genomics Viewer 2.11.2 [90], domain architecture was visualized using Jalview 2.11.2.5 [91]. Gene models with missing exons or exons erroneously predicted were fixed. Sequences of NLR candidates with frameshifts, premature stop codons, or domain architecture that did not fit the canonical architecture of NLRs were removed from the dataset. Furthermore, orthogroups that contained both NLR candidates and sequences for which no canonical NLR domain was predicted and sequences that did not prove similar to NLR candidates, at least for the NB domain, were also removed from the dataset. After this filtering step, NLR candidates in the 97 orthogroups included 168 sequences.

**Extending the set of NLR candidates in orthogroups with many NLRs.** Forty-seven orthogroups contained at least six NLR candidates. For each orthogroup, all annotated NB sequences were extracted, aligned using MAFFT 7.407 (default parameters), and visually inspected. For three orthogroups, two (OG0000039, OG0002684) or four (OG0007014) alignments were produced since the sequences included multiple NB domains. Each alignment for each orthogroup was used to build an HMM profile using HMMER 3.2.1. HMM profiles were then used for each orthogroup to search NB domains missed by the previous rounds of functional prediction using HMMER 3.2.1 (hmmsearch -E 1e-4). These HMM

profiles annotated an NB domain in 565 additional genes in the 47 orthogroups, adding to the 4943 NB domain genes already annotated. We also checked the domain content of all 47 orthogroups, leading to the exclusion of 10 orthogroups. OG0007227 was excluded because Ankyrin repeats were in N-terminal to the AAA domain, and because other domains (UBA_4, Clr5 and ClpB_D2-small) were identified. OG0007014, OG0003600, and OG0002684 were excluded because they appeared to belong to the gene family Structural Maintenance of Chromosomes based on their domain content and architecture. OG0005316, OG0004572, OG0001940, OG0000039, and OG0000358 were excluded because they appeared to belong to the gene families CaseinoLytic Peptidase B, two-component sensor protein histidine protein kinase (NCU01823 in *N. crassa*), septin, ABC transporter, and PHOsphatase D gene families, respectively, based on their domain content and architecture. OG0000027 was excluded because it lacks systematic C-terminal repeats, and other domains overlapped with the AAA (MMR_HSR1, RsgA_GTPase, Septin, AIG1 and Dynamin_N). After these filtering steps, the set of NLR candidates in the 37 remaining orthogroups included 5,599 sequences.

**Sequence correction and NLR prediction using EXONERATE.** Among the 5,599 NLR candidates, the 4,040 sequences longer than 1000 bp were aligned against the 84 genome assemblies using EXONERATE 2.4.0 (--model protein2genome --showtargetgff --maxintron 1000 --minintron 10). In parallel, we extracted all annotated NB domains from NLR candidate genes and aligned them against the genome assemblies using EXONERATE 2.4.0 with the same options. We noted the coordinates of all EXONERATE hits of NB domains, filtered out hits with stop codons and/or frameshifts, and combined sequences with overlapping coordinates. We recorded the coordinates of all EXONERATE hits of complete NLR sequences, filtered out hits with scores less than 200, filtered out hits whose coordinates did not overlap with EXONERATE hits of NB domains, filtered out hits with stop codons and/or frameshifts, and combined sequences with overlapping coordinates. By combining the results of the two sets of EXONERATE alignments, we could add new sequences to the set of NLR candidates and fix gene prediction errors, such as missing exons, erroneous introns, missing methionine, or missing stop codons, while ensuring that sequences possess an NB domain. After removing sequences shorter than 1000 bp and 282 Mitochondrial chaperones BCS1, we obtained a set of 4,613 *Sordariales* (S2 Table) and 153 outgroup NLR candidate sequences. The PYTHON script for this filtering is available on our github (https://github.com/bonospora/NLR_Sordariales).

## Orthology analysis

Our initial orthology analysis with ORTHOFINDER identified 47 NLR orthogroups from whole predicted proteomes. To summarize previous *Methods* sections, we then (1) fixed gene prediction errors, (2) removed non-NLR sequences, (3) added new NLR candidate sequences, (4) split orthogroups based on NB domain type. We then conducted a new orthology analysis by segregating NLRs according to their NB type.

**Clustering of NACHT and NB-ARC sequences.** All 4613 *Sordariales* NLR sequences were clustered independently for each NB type (NB-ARC, T-NACHT, and N-NACHT) and each family (*Sordariaceae*, *Podosporaceae*, *Lasiosphaeriaceae*, *Chaetomiaceae*) using ORTHOFINDER 2.5.2 (-M msa -S diamond_ultra_sens) with DIAMOND 2.0.9. Sequences from NLRs with no predicted NB were excluded from clustering at this stage.

For each combination of NB type × *Sordariales* family (excluding N-NACHT that formed a single orthogroup for each family), we ran ORTHOFINDER multiple times with varying values for the MCL inflation parameter (-I parameter). We tested the inflation parameter from 0.5 to 10 with a 0.5 increment. For each value of the inflation parameter, we counted the number of NLRs in the five largest NLR groups. If the number of NLRs remained the same for three consecutive values of the inflation parameter, we considered it a plateau. We set the inflation parameter to the value at which the plateau was reached. We finally used the following inflation parameter values: T-NACHT × *Chaetomiaceae*: 3.0; T-NACHT × *Lasiosphaeriaceae*: 5.0; T-NACHT × *Podosporaceae*: 3.0; T-NACHT × *Sordariaceae*: 5.0; NB-ARC × *Chaetomiaceae*: 5.0; NB-ARC × *Lasiosphaeriaceae*: 3.0; NB-ARC × *Podosporaceae*: 3.5; NB-ARC × *Sordariaceae*: 2.0 (S15 Table).

**Assignment of NLR sequences without predicted NB to T-NACHT, N-NACHT, and NB-ARC orthogroups.** We classified the NB domain of 60 NLR sequences that lacked an annotation by performing similarity searches with BLASTP

2.12.0 against sequences of other NLRs. We also analyzed NLRs that were part of small NLR orthogroups (five NLRs or less) to ensure they were not incorrectly annotated.

## Identification of NLR clusters

To determine whether certain NLRs formed clusters in *Sordariales* genomes, we first defined what is a fungal NLR cluster. NLR clusters in plants consist of at least two NLRs within a 200 kb window, separated by no more than eight non-NLR genes [35]. However, since fungal genomes are denser in genes than plant genomes, we tested various combinations of two criteria: the maximum distance between two NLRs (0, 0.5, 1, 1.5, 2, 2.5, 3, 3.5, 4, 4.5, 5, 6, 7, 8, 9, 10, 15, 20, 25, 30, 35, 40, 45, 50, 100, 150 or 200 kb) and the maximum number of non-NLR genes between two NLRs (0, 1, 2, 3, 4, 5, 6, 7 or 8). For each of the 37 genomes with an N50 higher than 1Mb, and for each combination of maximum inter-NLR distance and maximum number of inserted non-NLR, we determined the number of NLR genes identified as clustered and assessed the probability this number could be obtained by chance only. To do so, we randomly selected X genes, X being the number of NLRs in a given assembly, and counted how many would be identified as clustered based on the assumed maximum inter-NLR distance and maximum number of inserted non-NLR. We repeated this procedure 1,000 times. We finally compared the number of clustered NLRs to the mean number of clustered random genes. We selected parameter values 40 kb for the maximum distance between NLRs and eight for the maximum number of non-NLR genes as this maximized, for 80% of the assemblies, the difference between the number of clustered NLRs and the mean number of clustered genes.

## Association between NLRs and other variables

Phylogenetic generalized least squares (PGLS) analysis is a method to investigate the relationship between variables while accounting for the shared evolutionary history among species. We used phylogenetic generalized least squares analysis to examine the relationships between genome features, the number of NLR genes, and the number of NLR clusters. Genome features were the GC content, the repeat content, the RIP affected area, N50, L50, the number of contigs, the total length of the assembly, and the Busco score. GC-, RIP-, and repeat-related variables were obtained from Hensen *et al.* [79]. PGLS analyses were conducted using the CAPER and GEIGER packages in R.

We also compared the number of NLR genes and the number of NLR clusters to the bimodal distribution of GC content, also obtained from Hensen *et al.* [79]. Since the bimodal distribution of GC content is a binary parameter, we used the threshold model proposed by Felsenstein [92] implemented in the treshBayes function from the PHYTOOLS package in R [51]. For each analysis, we ran $2 \times 10^6$ generations, sampling every 1,000 generations, we removed half of the chain as burn-in, and we determined the highest posterior density interval of 95% probability.

## Functional annotation of NLRs

Having identified 4613 NLR genes in *Sordariales*, we functionally annotated them using built-in Pfam-A (version 33.1) HMM profiles, our *Sordariales*-specific HMM profiles (see previous *Methods* sections), and HMM profiles from reference [16], referred to as Wojciechowski profiles. HMMER 3.2.1 was run on each of the 82 sets of Sordariales NLRs (hmmsearch -E 1e-4) with all three HMM profiles.

To annotate NB domains, we only used *Sordariales*-specific HMM profiles, as Wojciechowski profiles do not cover NB domains. Three situations were encountered: (1) for 60 sequences, no NB domain was predicted; (2) for 3909 sequences, one and only one NB domain type was identified; (3) for 644 sequences, two or three different, overlapping, NB domains were predicted. For the 644 sequences with multiple overlapping NB domains predicted, we retained the domain for which the lowest E-value (the "best" annotation) was lower than the squared E-values of other competing predictions, or we considered that the NB domain was present but undetermined.

Functional annotation of other C-terminal and N-terminal domains was performed using the same approach as with the NB domain, using built-in Pfam-A, *Sordariales*-specific and Wojciechowski HMM profiles. If different overlapping domains were predicted with different HMM profiles, we prioritized predictions with Wojciechowski profiles over predictions with *Sordariales*-specific profiles, and prediction with *Sordariales*-specific profiles over predictions with Pfam-A. If different overlapping domains were predicted with the same HMM set of profiles, we retained the domain for which the lowest E-value (the "best" annotation) was lower than the squared E-values of other competing predictions, otherwise all predictions were kept.

We considered canonical the C-terminal domains in categories Ankyrin repeats, TPR, WD40 repeats, or HEAT repeats. We considered canonical the N-terminal domains in categories HeLo-like, sesB-like, Goodbye-like, HET-like, PNP_UDP_1, RelA_SpoT and Patatin. We merged categories HeLo-like domains and sesA, as well as NAD1 and Goodbye because, as noticed in Dyrka *et al.* [2], we observed extensive overlap between the sesA and HeLo-like annotated sets and the NAD1 and Goodbye annotated sets. We also grouped all amyloid motif domains under the "amyloid" category. All the canonical and amyloid N-terminal domains are listed in S15 Table. All other domains were categorized as non-canonical domains.

### HMM profiles for *Sordariales* T-NACHT, NB-ARC, and N-NACHT domains

Having discovered that the fungal NACHT domain could be subdivided into two sub-types (T-NACHT and N-NACHT; see *Results*), we built improved versions of our *Sordariales*-specific HMM profiles. For each type of NB, NLR sequences were aligned using muscle 5.1 (default settings), and only the NB domain was kept. The resulting 2258 T-NACHT, 839 NB-ARC, and 45 N-NACHT sequences were re-aligned using muscle 5.1 (-super5). The alignments were used to build HMM profiles with HMMER 3.2.1 (hmmbuild --amino --informat afa).

### NB alignment-free sequences comparison

We relied on alignment-free approaches to compare T-NACHT, NB-ARC, and N-NACHT sequences. We randomly sampled one NB domain per NLR orthogroup. The resulting 325 T-NACHT, 85 NB-ARC, and four N-NACHT were split into two fragments, between the Walker A and Walker B motifs (Fig 3A). Alignment-free distances were computed using the Alfree package [57]. Principal Coordinates Analysis (PCoA) was performed using the cmdscale function in R.

### Sequence comparisons between PKinase domains in NLRs and non-NLRs

We identified a set of nine NLRs in *Podosporaceae* and *Chaetomiaceae* that all included a C-terminal PKinase domain. To investigate the evolutionary origin of these NLR sequences and test the hypothesis of a single integration of the PKinase domain, we inferred the genealogy of the NB domain of all 40 sequences belonging to the two corresponding NLR orthogroups (11 *Podosporaceae* sequences, 29 *Chaetomiaceae* sequences). NB sequences were aligned with MUSCLE 3.8.31, and their genealogy was inferred using RAxML-ng 0.9.0 with the following commands: --all --model GTR+G --bs-trees 100 --tree pars [80],rand [80]. In parallel, PKinase sequences were used for similarity searches using BLASTP (blast 2.9.0; blastp -max_target_seqs 2000 -threshold 300) against sequences of all *Sordariales* proteomes. Having identified high similarity between PKinase sequences of the nine NLRs and other PKinase sequences in non-NLR genes belonging to orthogroup OG0001798, we compared PKinase sequences from NLRs and non-NLRs. We aligned sequences in JALVIEW 2.11.2.5 using MUSCLE 3.8.31 (default settings), and we calculated the BLOSUM62 distance index between pairwise sequences using the DistanceCalculator function from the PYTHON package BIO.PHYLO.TREECONSTRUCTION.

### Identification of N-NACHT NLRs in Fungi

To investigate the distribution of NLRs with an N-NACHT NB domain across the fungal kingdom, we assembled a dataset of 122 freely available proteomes representative of the whole fungal kingdom, except *Sordariales*, and we selected

one proteome per fungal order reported in James *et al.* [58]. Proteomes were obtained from the NCBI (National Center for Biotechnology Information), the JGI (Joint genome Institute) or the supplementary materials from published articles (S8 Table). The dataset included two *Rozellomycota*, three *Blastocladiomycota*, ten *Chytridiomycota*, one *Olpidiales*, six *Zoopagomycota*, eight *Mucoromycota*, 39 *Basidiomycota* and 53 *Ascomycota*. We used Pfam-A and *Sordariales*-specific HMM profiles in HMMER 3.2.1 (hmmsearch -E 1e-4) to identify NLRs as genes harboring a least a NB domain and a canonical C-terminal domain. Leucine-Rich Repeats were included in C canonical domain for this analysis (Pfam-A HMM profiles: LRR19-TM, LRRC37, LRRC37AB_C, LRRCT, LRRFIP, LRRNT, LRRNT_2, LRR_adjacent, LRR_RI_capping, NEL, TTSSLRR, CENP-F_leu_zip, LRV, LURAP, and LRR_k with k in [1–6,8–12]).

## Supporting information

**S1 Fig. Comparison of species trees built using the Astral program (left) and Maximum-likelihood phylogenetic inference based on concatenated data (right).** Topological differences are highlighted in purple, with the corresponding leaves connected by red lines. Astral phylogeny was built using 2367 gene trees from single-copy orthologs including at least 80% of species. Maximum-likelihood phylogeny was built using 1030 single-copy core orthologous sequences. (DOCX)

**S2 Fig. Principal Coordinates Analysis of distance matrices at the first and second fragments of the NB domain.** The analysis was conducted with eight different modalities. Distances were computed using eight different alignment-free methods: Base-Base Correlation index (maximum distance to observe a correlation between bases set to 10), Normalized Compression Distance, W-metric (with a BLOSUM62 matrix), d2 distance (with the minimum and maximum word sizes set respectively to 2 and 3 and the vector set to frequencies), Composition Distance (with word size set to 3), Normalized Google Distance (with word size set to 2 and vector set to frequencies), Normalized Google Distance (with word size set to 4 and vector set to frequencies), Sorensen-Dice Coefficient index (with word size set to 3). (DOCX)

**S3 Fig. Phylogeny of all NACHT domains in orthogroups OGpodoTNACHT005 and OGchaetoTNACHT008.** Branches corresponding to NLR sequences containing a Pkinase domain are highlighted in blue (Podospora bulbillosa), orange (Corynascus sp.), or red (other Chaetomiaceae species). Phylogeny was inferred using RAXML-NG on NACHT sequences aligned using MUSCLE. Nodes with a bootstrap value higher than 60% are highlighted with a black dot. (DOCX)

**S1 Table. Eighty-two Sordariales genomes and two outgroup genomes included in the study, with their taxonomic classification, isolate metadata, genome features, number of NLR genes, number of NLR clusters, and number of clustered NLRs.** (XLSX)

**S2 Table. Orthogroups identified from the orthology analysis of 84 Sordariales genomes.** (XLSX)

**S3 Table. Functional annotations of the 4613 NLRs identified in the 82 Sordariales genomes (amino acid positions between brackets).** (XLSX)

**S4 Table. Orthogroups identified from the orthology analysis of 4613 NLR sequences in Sordariales.** (XLSX)

**S5 Table. Results of Phylogenetic Generalised Least Squares (PGLS) analysis of the relationship between the number of NLR genes or NLR clusters, and several genome statistics. For each analysis, we provide the lambda**

value alone and in relationship with the number of NLR genes or clusters. We also provide the strength of the effect of the index on the number of NLR genes or clusters (slope PGLS), the related adjusted R², and the p-value of the association between each statistic and the number of NLR genes.
(XLSX)

S6 Table. Thirteen head-to-head paired NLRs identified in Sordariales. The names and the functional annotations are provided.
(XLSX)

S7 Table. Genomes representative of the whole fungal kingdom (Sordariales excluded) used for identifying the phylogenetic distribution of N-NACHT NLRs in fungi. The taxonomic affiliation of each genome is provided.
(XLSX)

S8 Table. NLRs identified in 122 proteomes representative of the whole fungal kingdom. For each NLR gene identified, we provide its genome of origin, gene name, type of NB domain (T-NACHT, NB-ARC, N-NACHT or undetermined), and identity of other domains identified. All genes containing a canonical NB domain and a canonical C-terminal domain were identified as NLRs. N-NACHT NLRs are highlighted in blue, NB-ARC NLRs are in yellow, N-NACHT NLRs are in red, and NLRs with an undetermined NB domain are in white. Genes with both a canonical NB domain and a canonical C-terminal domain but whose architecture was incompatible with being identified as NLRs were not included in the set NLRs (such as SMC proteins) and are highlighted in dark grey.
(XLSX)

S9 Table. Distribution of NLR domain assortments in Sordariales.
(XLSX)

S10 Table. Integrated non-canonical domains identified in Sordariales NLRs and the corresponding number of NLRs in which they were found.
(XLSX)

S11 Table. BLOSUM62 distances between protein kinase domains from NLRs in orthogroups OGpodoT-NACHT005 and OGchaetoTNACHT008 and protein kinase domains from CHK-2 proteins in Sordariales.
(XLSX)

S12 Table. Pfam-A domains retained to build Sordariales-specific HMM profiles and for NLR identification. These domains correspond to canonical NB and C-terminal domains of NLRs, or to domains that could be found as repeats in NLRs based on Dyrka et al. 2014 [1].
(XLSX)

S13 Table. RNAseq data used for the manual correction of NLRs.
(XLSX)

S14 Table. Variation in the number of NLRs in the five largest orthogroups of NLRs, for each NLR orthology analysis (family x type of NB), as a function of the inflation parameter (I). The value for the inflation parameter retained in each NLR orthology analysis is highlighted.
(XLSX)

S15 Table. Domains retained for the canonical N-terminal annotation of NLRs in Sordariales, with the category they belong to (HeLo-like, sesB-like, Goodbye-like, HET-like, PNP_UDP_1, RelA_SpoT, Patatin, or amyloid), and the model used to identify them (Wojciechowski et al. 2022 [1], Pfam-A, or Sordariales-specific).
(XLSX)

## Author contributions

**Conceptualization:** Lucas Bonometti, Silvia Miñana-Posada, Pierre Gladieux.

**Data curation:** Lucas Bonometti, Silvia Miñana-Posada, Pierre Gladieux.

**Formal analysis:** Lucas Bonometti, Noah Hensen, Silvia Miñana-Posada, Pierre Gladieux.

**Funding acquisition:** Hanna Johannesson, Pierre Gladieux.

**Investigation:** Lucas Bonometti, Noah Hensen, Silvia Miñana-Posada, Pierre Gladieux.

**Methodology:** Lucas Bonometti, Florian Charriat, Noah Hensen, Pierre Gladieux.

**Project administration:** Hanna Johannesson, Pierre Gladieux.

**Software:** Lucas Bonometti, Florian Charriat.

**Supervision:** Hanna Johannesson, Pierre Gladieux.

**Visualization:** Lucas Bonometti.

**Writing – original draft:** Lucas Bonometti, Pierre Gladieux.

**Writing – review & editing:** Lucas Bonometti, Noah Hensen, Hanna Johannesson, Pierre Gladieux.

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
