## [Decision Letter · Decision Letter 0]

PGENETICS-D-24-01497

Systematic exploration of domain assortments in NOD-like receptors uncovers two types of NACHT domains in Sordariales fungi

PLOS Genetics

Dear Dr. Gladieux,

Thank you for submitting your manuscript to PLOS Genetics. After careful consideration, we feel that it has merit but does not fully meet PLOS Genetics's publication criteria as it currently stands. Therefore, we invite you to submit a revised version of the manuscript that addresses the points raised during the review process.

Please submit your revised manuscript within 30 days Mar 14 2025 11:59PM. If you will need more time than this to complete your revisions, please reply to this message or contact the journal office at plosgenetics@plos.org. Please include the following items when submitting your revised manuscript:

We look forward to receiving your revised manuscript.

Kind regards,

Benjamin Schwessinger

Academic Editor

PLOS Genetics

Geraldine Butler

Section Editor

PLOS Genetics

Aimée Dudley

Editor-in-Chief

PLOS Genetics

Anne Goriely

Editor-in-Chief

PLOS Genetics

**Journal Requirements:**

At this stage, the following Authors/Authors require contributions: Lucas Bonometti, Florian Charriat, Noah Hensen, Silvia Miñana-Posada, Hanna Johannesson, and Pierre Gladieux. Please ensure that the full contributions of each author are acknowledged in the "Add/Edit/Remove Authors" section of our submission form.

The list of CRediT author contributions may be found here: https://journals.plos.org/plosgenetics/s/authorship#loc-author-contributions

Potential Copyright Issues:

i) Figure 1. Please confirm whether you drew the images / clip-art within the figure panels by hand. If you did not draw the images, please provide (a) a link to the source of the images or icons and their license / terms of use; or (b) written permission from the copyright holder to publish the images or icons under our CC BY 4.0 license. Alternatively, you may replace the images with open source alternatives. See these open source resources you may use to replace images / clip-art:

5) Thank you for stating that “Genome assemblies, gene models, HMM profiles, orthogroups and NLR sequences generated in this study are available from Zenodo (doi: 10.5281/zenodo.14204294).” We noted that  the data files are restricted to users with access. Please assure that the datasets are publicly accessible.

**Reviewers' comments:**

Reviewer's Responses to Questions

Reviewer #1: In this study, the authors performed a systematic exploration of NLR diversity in the Sordariales order of Ascomycota fungi. They improved the annotation of the NB domain, and found a high variation in the number of NLR genes among species. They found NLR were organized in clusters based on their definition of clusters in these fungi.

Overall, I found this study highly novel in term of approach and methodology that should be of interest for scientific community interested in the evolution of immune systems and fungi. I haven't detected major methodological flows. The authors tackle putative biais due to differences in genome quality in their dataset and use both automatic and trained annotation pipelines accompanied with manual curation steps.

I have only a few minor remarks:

-L110: it is unclear to me what means "the determinants of NLR"

-the author claims the Sordariales are good systems to study the link between ecology and genome evolution (L128) but conclude in the discussion that there is not enough knowledge on Sordariales lifestyle (L507); This is contradictory and I wonder if the authors could have tried to correlate substrate of origin of strains and their NLR repertoire for examples

-the author conclude the fungi share many similarity in NRL repertoire to animals (L603), but very little information on NLR repertoire in animals is given ; most comparison are done with plants, in particular for clusters. Is there a specific reason for this?

-L145 : a N50 of 14.1 kb seems very low and it seems unlikely that many genes can be annotated in this genome

-L393 and beyond: I failed understanding the main results of this analysis ; L420 the author claims "only a subset of possible combinations was observed in Sordariales" and conclude L426 that the "realized proportion of possible domain assortments were only marginally lower in Sordariales than in Dikarya"; the two sentences seems contradictory to me

Reviewer #2: In their manuscript, ‘Systemic exploration of domain assortments of NOD-like receptors uncovers two type of NACHT domains in Sordariales fungi’, Bonometti et al. characterize in silico the NLR gene repertoire, genomic organization, and diversity in the fungal order of Sordariales. They authors identify and analyze 4613 NLR genes in 82 Sordariales genomes, uncover and report several intriguing findings about fungal NLRs in Sordariales, including the presence of NLR gene clusters, and the presence of two main types of NACHT domains in fungi. These findings, as well as the reported association of number of non-canonical C-terminal domains, which are hypothesized to represent ‘integrated NLR decoys’ in Sordariales, represent exciting contributions to NLR biology in general, and its exploration in fungi in particular. In addition, the presented work is methodologically sound and the manuscript mostly well-written. Below, I have some comments and suggestions that would, in my view, ameliorate the manuscript.

1) Considering the multiple discoveries and the broad scope of the analyses, the title appears to be somehow limiting into presenting the overall impact of the paper. An alternative could perhaps appear to better reflect the whole and also more appealing for a boarder readership.

2) In this context, I think that two other findings ‘NLR clustering’ and ‘Accessory C-ter NLR domains’ could benefit from additional illustrations. For example, cartoons showing some examples of such clusters, NLRs as illustrations. Plus supplementary Tables where one could easily find each such gene from these findings, with species names, and gene ID.

3) Regarding the NLR gene clusters in fungi and their definition/discovery. In my view, the text could be slightly clearer regarding the explanations of their identification (the phrase lines 258-261). Intuitively, one would think that clusters inside the same gene family would depend on the number of gene members and the genome size and gene density. To establish if clustering of gene members exist, one would need to determine if associations between members is more frequently found than what one would expect purely by chance. The description doesn’t provide enough context in my view to make this clear. Are the 1000 randomly-selected gene sets members of similarly sized (as the NLRs) gene families?

What could one expect to find in fungi, if we simply take the plant definition and ‘scale it down’ to account for the average differences in genome sizes, and distance between genes, and average size of the number of genes inside the plant and fungal NLRomes? Wouldn’t another option be to take such parameters from plants and attempt to apply the definition of NLR clusters from there, adapting it for fungi? How would such approach differ from what the authors propose?

4) It feels like the section regarding the identified orthogroups could be presented with a little more details. What are the NLR architectures in the largest orthogroups? Which species are concerned? One could hypothesize that largest orthogroups play important roles and potentially have conserved function in all Sordariales species, so it would be a valuable information for the readers. What percentage of NLRs in Sordariales do not fall in any of the orthogroups? In general, adding some % in this section would be beneficial for the article, especially for future experimental analysis of some of these NLRs.

Some additional minor comments below:

5) The definition of ‘head-to-head NLRs’ and its implication, I think, is only provided in the Discussion section. That could be provided earlier, when the term is initially mentioned.

6) The very first line of the Abstract, the NLRs in fungi are defined as ‘analogs’. Considering the presence of NLRs in bacteria, and their implication in immunity as anti-phage defense systems, and that the fungal NLRs paly similar role in immune defense in the context of allorecognition, it would seem justified to consider NLRs evolutionary related, originating from an ancient NB (NOD) domain, and thus distant homologs.

**Have all data underlying the figures and results presented in the manuscript been provided?**

Reviewer #1: Yes

Reviewer #2: None

PLOS authors have the option to publish the peer review history of their article (what does this mean? ). If published, this will include your full peer review and any attached files.

**Do you want your identity to be public for this peer review?** For information about this choice, including consent withdrawal, please see our Privacy Policy .

Reviewer #1: No

Reviewer #2: No

**Figure resubmission:**
---

## [Editor Report · Decision Letter 1]

Dear Dr Gladieux,

We are pleased to inform you that your manuscript entitled "Genomic organization, domain assortments, and nucleotide-binding domain diversity of NLR proteins in Sordariales fungi" has been editorially accepted for publication in PLOS Genetics. Congratulations!

Yours sincerely,

Benjamin Schwessinger

Academic Editor

PLOS Genetics

Geraldine Butler

Section Editor

PLOS Genetics

Aimée Dudley

Editor-in-Chief

PLOS Genetics

Anne Goriely

Editor-in-Chief

PLOS Genetics

Comments from the reviewers (if applicable):

**Data Deposition**

http://datadryad.org/submit?journalID=pgenetics&manu=PGENETICS-D-24-01497R1

**Press Queries**

---

## [Editor Report · Acceptance letter]

PGENETICS-D-24-01497R1

Genomic organization, domain assortments, and nucleotide-binding domain diversity of NLR proteins in Sordariales fungi

Dear Dr Gladieux,

We are pleased to inform you that your manuscript entitled "Genomic organization, domain assortments, and nucleotide-binding domain diversity of NLR proteins in Sordariales fungi" has been formally accepted for publication in PLOS Genetics! Your manuscript is now with our production department and you will be notified of the publication date in due course.

With kind regards,

Anita Estes

PLOS Genetics

On behalf of:
